# Isospin asymmetry in holographic baryonic matter

**Nicolas Kovensky[1,2]⋆ and Andreas Schmitt[1]†**

**1** Mathematical Sciences and STAG Research Centre, University of Southampton,
Southampton SO17 1BJ, United Kingdom
**2** Institut de Physique Théorique, Université Paris Saclay, CEA, CNRS,
Orme des Merisiers, 91191 Gif-sur-Yvette CEDEX, France

⋆ nicolas.kovensky@ipht.fr,  † a.schmitt@soton.ac.uk

## Abstract

We study baryonic matter with isospin asymmetry, including fully dynamically its interplay with pion condensation. To this end, we employ the holographic Witten-Sakai-Sugimoto model and the so-called homogeneous ansatz for the gauge fields in the bulk to describe baryonic matter. Within the confined geometry and restricting ourselves to the chiral limit, we map out the phase structure in the presence of baryon and isospin chemical potentials, showing that for sufficiently large chemical potentials condensed pions and isospin-asymmetric baryonic matter coexist. We also present first results of the same approach in the deconfined geometry and demonstrate that this case, albeit technically more involved, is better suited for comparisons with and predictions for real-world QCD. Our study lays the ground for future improved holographic studies aiming towards a realistic description of charge neutral, beta-equilibrated matter in compact stars, and also for more refined comparisons with lattice studies at nonzero isospin chemical potential.

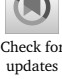
# 1   Introduction

## 1.1   Motivation

Dense nuclear matter in compact stars contains more neutrons than protons due to the conditions of electric charge neutrality and equilibrium with respect to the weak nuclear force. This isospin asymmetry is routinely taken into account in effective field theories and phenomenological models of nuclear matter and applications thereof to the physics of compact stars [1]. In recent years, the gauge-gravity duality [2–4] has been increasingly applied to dense matter as well, providing a rigorous strong-coupling approach, albeit in theories that differ more or less from Quantum Chromodynamics (QCD), the relevant underlying theory. These studies either focus on a holographic version of quark matter, to be combined with a field-theoretical description of nuclear matter if applied to compact stars [5–8], or they employ isospin-symmetric nuclear matter for simplicity [9–15]. In this paper we develop a more realistic approach by including an isospin asymmetry into the holographic description of dense baryonic matter.

Besides the phenomenology of compact stars, our motivation can also be put into a more general theoretical context. While under compact star conditions the isospin asymmetry adjusts itself dynamically for a given baryon density and temperature, for a more general treatment one may consider the isospin chemical potential $\mu_I$ as an independent thermodynamic variable. At the fundamental level, $\mu_I$ introduces an imbalance between $u$ and $d$ quarks, and one may investigate the QCD phase structure in the space spanned by $\mu_I$, baryon chemical potential $\mu_B$, and temperature $T$. For $\mu_B = 0$, brute force lattice calculations can be employed because $\mu_I$ on its own does not induce a so-called sign problem [16–20]. As suggested by chiral perturbation theory [21–27], lattice QCD confirms that Bose-Einstein condensation of charged pions sets in when $\mu_I$ becomes larger than (half of) the pion mass. As $\mu_I$ is increased further, eventually a deconfined regime is reached, where perturbative methods become applicable [22, 28, 29]. It was conjectured that the zero-temperature transition from the pion-condensed phase to the deconfined phase at ultra-high isospin density is smooth, in particular

without the appearance of baryonic degrees of freedom [21]. Our holographic approach allows us to investigate the phase structure of the model for arbitrary $\mu_I$, $\mu_B$, and $T$. In particular we allow for pion condensation and baryonic matter and their coexistence, and we determine the preferred phase fully dynamically – in fact, we shall find that baryons do appear even at infinitesimally small $\mu_B$ if $\mu_I$ is sufficiently large.

## 1.2 Model

The gauge-gravity duality provides a window into the strongly coupled regime of QCD-like theories with a large number of colors $N_c$, where the relevant observables can be studied by means of classical gravity computations. The holographic dual of QCD is currently not known. Perhaps closest to real-world QCD is the Witten-Sakai-Sugimoto model [30–32]; for a review see Ref. [33]. At weak 't Hooft coupling $\lambda$ and low energies, i.e., below the Kaluza-Klein scale $M_{\mathrm{KK}}$ induced by a compactified extra dimension, it is dual to large-$N_c$ QCD. However, the gravitational description is applicable only in the opposite regime, where $\lambda$ becomes large and the curvature of the background is small. Nevertheless, if interpreted with some care, the model proves to be very useful to capture non-perturbative QCD-like effects, which are otherwise very difficult to obtain from field-theoretical approaches. The model has been employed to compute spectrum and couplings of mesons [31, 32], properties of glueballs [34, 35], and static properties of nucleons [36, 37]. It was soon realized that it can also be employed to study thermodynamic phases and the phase transitions between them [10, 38, 39]. The physics of the gluons is captured by a gravitational background generated by $N_c$ D4-branes, which give rise to two different geometries, interpreted as confined and deconfined phases and separated by the critical temperature $T_c = M_{\mathrm{KK}}/(2\pi)$ [30]. Flavor degrees of freedom are included via $N_f$ D8- and $\overline{\mathrm{D8}}$-branes, accounting for left- and right-handed fermions, and spontaneous breaking of chiral symmetry is geometrically realized by a configuration where branes and anti-branes connect in the bulk [31, 32]. The scale associated with chiral symmetry breaking is set by the asymptotic separation of the flavor branes $L$ and can be decoupled from the deconfinement scale.

We will start our study within the confined geometry and maximal brane separation (on antipodal points of the compactified extra dimension with radius $M_{\mathrm{KK}}^{-1}$). This is the simplest version of the model and allows us to explain our setup in a transparent way and to evaluate the different phases numerically without difficulties. We also extend this approach to the deconfined geometry, where the flavor brane configuration has to be calculated dynamically. It has been argued that this setup, in particular its "decompactified limit", is better suited to capture features of real-world QCD, at least with respect to the chiral phase transition [9, 11, 39]. In our context, the calculation in the deconfined geometry turns out to be numerically challenging, and we shall only present some selected results, leaving a more systematic evaluation for the future. In both scenarios we shall work with two flavors, $N_f = 2$, and employ the probe brane limit, $N_f \ll N_c$, where the backreaction of the flavor branes onto the background geometry is neglected (see Refs. [40–42] and [13, 43] for attempts to incorporate these effects in the Witten-Sakai-Sugimoto model and within the so-called holographic V-QCD model, respectively).

## 1.3 Method and approximations

The main focus of our study is baryonic matter. Baryons are intrinsically heavy objects in the 't Hooft limit since their masses grow linearly with $N_c$. In holography, they are realized as solitonic D-branes wrapping compact directions [44] or equivalently as instanton configurations of the gauge fields in the bulk [45]. Generalizing these single-baryon configurations to many-baryon systems is extremely complicated, and various approximations have been employed

in the literature. In the Witten-Sakai-Sugimoto model, a superposition of pointlike baryons has been considered as a model for homogeneous nuclear matter [12, 46]. Improvements of this approach are based on the single-instanton solution – which for two flavors and in the flat-space limit is given by the well-known Belavin-Polyakov-Schwarz-Tyupkin (BPST) instanton [47] – to construct an instanton gas [9, 10], further refined by using the two-instanton solution to incorporate two-body interactions [11]. In this framework also crystalline phases of holographic nuclear matter were studied [48, 49]. Here we are only interested in homogeneous phases and will start from an ansatz for the non-abelian gauge fields on the flavor branes that is homogeneous in the spatial directions of the field theory. This approach is somewhat complementary to the instanton approach and is expected to be valid at large baryon densities. It was pioneered in Ref. [50] and improved in different ways in Refs. [9, 51] (it has also been used in Ref. [13] in the V-QCD setup). Our calculation can be viewed as a generalization of the homogeneous ansatz for baryonic matter of Ref. [9] to nonzero $\mu_I$.

One can also view our calculation as a generalization of purely mesonic studies within the Witten-Sakai-Sugimoto model at nonzero $\mu_I$ by adding baryons. In the absence of baryons, pion condensation can be included by choosing the boundary conditions for the gauge fields on the flavor branes appropriately [52, 53]. Previous studies in this context have been performed in the chiral limit, i.e., in the absence of current quark masses and thus at zero pion mass. Therefore, pion condensation sets in as soon as $\mu_I$ is nonzero (the configuration is destabilized by rho meson condensation at large $\mu_I$ [52]). Including current quark masses is not straightforward in the Witten-Sakai-Sugimoto model. For small masses this can be done in an effective way [54–59], and a consistent evaluation of the phase diagram for nonzero pion mass is possible [12, 60]. Nevertheless, this effect would complicate our calculation significantly and thus we shall restrict ourselves to the chiral limit in this paper. As a consequence, we will find that in the energetically preferred phases baryonic matter is always accompanied by pion condensation. We also construct the configuration for isospin polarized baryons without pion condensation, anticipating that this phase will be preferred in certain regions of the phase diagram once a nonzero pion mass is included.

Within our holographic approach and the given approximations it is unavoidable that large-$N_c$ properties of isospin-asymmetric baryonic matter will be manifest and comparisons to $N_c = 3$ QCD have to be taken with care. Most importantly, the energy differences between baryonic states with different isospin values go to zero as $N_c \to \infty$, such that the spectrum becomes continuous with respect to isospin. Within the Witten-Sakai-Sugimoto model, this spectrum was calculated and it was shown that quantization in the bulk gives a discrete spectrum where neutron and proton states can be identified [36] (the neutron/proton mass difference can be calculated as well [61]). Our homogeneous ansatz does not include this quantization, and thus in particular there are baryons with zero isospin number, from which isospin symmetric matter can be created. This is different from ordinary symmetric nuclear matter, which is made of an equal number of neutrons and protons. In that case, isospin asymmetry can be created by rearranging the Fermi surfaces of neutrons and protons and as a result the symmetry energy is more than an order of magnitude smaller than the nucleon mass. We shall see that our approach yields a much larger symmetry energy since creating isospin polarized baryonic matter requires populating heavier states. Possibly to be explored in combination with our current approach in the future, one could attempt to construct a holographic many-body system of neutrons and protons explicitly. A simple version of such a construction was discussed in a setup similar to the Witten-Sakai-Sugimoto model, assuming that protons and neutrons at large $N_c$ consist of $N_c/3$ copies of $uud$ and $ddu$, which indeed leads to a symmetry energy comparable to ordinary nuclear matter [62, 63].

## 1.4  Outline of the paper

Our paper is organized as follows. In Sec. 2 we develop our formalism within the confined geometry with antipodal brane separation. This includes the setup in Sec. 2.1, a brief discussion of the single-baryon spectrum in Sec. 2.2, our ansatz for the gauge fields and their boundary conditions in Secs. 2.3 and 2.4, the free energy density and candidate phases in Secs. 2.5 and 2.6, and the low-density approximation, for which analytical results can be obtained, in Sec. 2.7. The numerical results of the confined geometry are presented and discussed in Sec. 3. Section 4 is devoted to the deconfined geometry, with the derivations in Secs. 4.1-4.3 similar to but technically more involved than for the confined case. Numerical results for the phase diagram and the onset of baryonic matter are discussed in Sec. 4.4. We summarize and give an outlook in Sec. 5.

## 2  Confined geometry

### 2.1  Setup

We start with the simplest version of the Witten-Sakai-Sugimoto model as constructed in the original works [30, 31]. The background geometry is sourced by $N_c$ D4-branes. One of their transverse directions, say $X_4$, is compactified on a circle with radius $M_{\rm KK}^{-1}$, and the chosen periodicity conditions break supersymmetry. At the lowest order in $N_f/N_c$, adding $N_f$ pairs of D8-$\overline{\rm D8}$-branes on this fixed background corresponds to including $N_f$ flavors of left- and right-handed fundamental quarks. They are located at the antipodes of the $X_4$ circle, such that their asymptotic separation is $L = \pi/M_{\rm KK}$. In this section we consider the confined geometry, where the subspace spanned by the holographic coordinate $U$ and $X_4$ is cigar-shaped with its tip at $U = U_{\rm KK}$, where $U_{\rm KK} = 2\lambda M_{\rm KK}\ell_s^2/9$, with $\lambda$ the 't Hooft coupling and $\ell_s$ the string length. In the confined geometry and with antipodal separation at $U = \infty$, the flavor branes are forced to join in the bulk at $U = U_{\rm KK}$, which is a realization of spontaneous chiral symmetry breaking in the IR according to the symmetry breaking pattern $U(N_f)_L \times U(N_f)_R \to U(N_f)_{L+R}$.

The metric on the flavor branes is given by

$$ds^2 = \left(\frac{U}{R}\right)^{3/2}\left(dX_0^2 + d\boldsymbol{X}^2\right) + \left(\frac{R}{U}\right)^{3/2}\left[\frac{dU^2}{f(U)} + U^2 d\Omega_4^2\right],\tag{1}$$

where $d\Omega_4^2$ is the metric of a unit 4-sphere, $R$ is the background curvature, related to the model parameters by $R^3 = \lambda\ell_s^2/(2M_{\rm KK})$, and

$$f(U) = 1 - \frac{U_{\rm KK}^3}{U^3}.\tag{2}$$

We work in Euclidean spacetime $(X_0, \boldsymbol{X})$ with Euclidean time $X_0 \in [0, 1/T]$. In this section, the temperature $T$ plays no role since the thermodynamic potential will turn out to be independent of $T$. This is different in the deconfined geometry, see Sec. 4, where nontrivial temperature effects become important.

The action on the flavor branes has a Dirac-Born-Infeld (DBI) and a Chern-Simons (CS) part,

$$S = S_{\rm DBI} + S_{\rm CS}.\tag{3}$$

Here, the DBI action is

$$S_{\rm DBI} = 2T_8 V_4 \int d^4 X \int_{U_{\rm KK}}^{\infty} dU e^{-\phi}\, {\rm STr}\sqrt{\det(g + 2\pi\alpha'\mathcal{F})},\tag{4}$$

with the D8-brane tension $T_8 = 1/\left[(2\pi)^8 \ell_s^9\right]$, the volume of the 4-sphere $V_4 = 8\pi^2/3$, the dilaton given by $e^\phi = g_s(U/R)^{3/4}$ with the string coupling $g_s = \lambda/(2\pi N_c M_{\rm KK} \ell_s)$, and $\alpha' = \ell_s^2$. Moreover, $g$ is the metric given by Eq. (1) and $\mathcal{F}$ is the field strength of the world-volume gauge field $\mathcal{A}$,

$$\mathcal{F}_{\mu\nu} = \partial_\mu \mathcal{A}_\nu - \partial_\nu \mathcal{A}_\mu + i[\mathcal{A}_\mu, \mathcal{A}_\nu], \tag{5}$$

with $\mu, \nu \in \{0, 1, 2, 3, U\}$. The factor 2 in Eq. (4) accounts for the two halves of the flavor branes. Since we are interested in the non-abelian case $N_f = 2$, a prescription for computing the square root is required in general. We have indicated this in the notation by including the symmetrized trace "STr" in Eq. (4). We shall comment on this prescription in more detail in Sec. 4 and continue here with the Yang-Mills (YM) approximation, where we can compute the determinant in Eq. (4) as if the gauge fields were abelian, expand the result up to order $\mathcal{F}^2$, and then take the ordinary trace. To this order, the result is identical with the symmetrized trace prescription.

We decompose the $U(2)$ gauge fields into $U(1)$ and $SU(2)$ parts,

$$\mathcal{A}_\mu = \hat{A}_\mu + A_\mu, \qquad A_\mu = A_\mu^a \sigma_a, \tag{6}$$

with the Pauli matrices $\sigma_a$, $a = 1, 2, 3$, normalized such that $\text{Tr}[\sigma_a \sigma_b] = 2\delta_{ab}$ and $[\sigma_a, \sigma_b] = 2i\epsilon_{abc}\sigma_c$, and analogously for the field strengths,

$$\mathcal{F}_{\mu\nu} = \hat{F}_{\mu\nu} + F_{\mu\nu}, \qquad F_{\mu\nu} = F_{\mu\nu}^a \sigma_a, \tag{7}$$

where

$$\hat{F}_{\mu\nu} = \partial_\mu \hat{A}_\nu - \partial_\nu \hat{A}_\mu, \qquad F_{\mu\nu}^a = \partial_\mu A_\nu^a - \partial_\nu A_\mu^a - 2\epsilon_{abc} A_\mu^b A_\nu^c. \tag{8}$$

The CS action can be written in terms of abelian and non-abelian components as [36, 53]

$$
\begin{aligned}
S_{\rm CS} = & -i\frac{N_c}{12\pi^2} \int d^4 X \int_{U_{\rm KK}}^\infty dU \left\{ \frac{3}{2}\hat{A}_\mu \left( F_{\nu\rho}^a F_{\sigma\lambda}^a + \frac{1}{3}\hat{F}_{\nu\rho}\hat{F}_{\sigma\lambda} \right) \right. \\
& \left. + 2\,\partial_\mu \left[ \hat{A}_\nu \left( F_{\rho\sigma}^a A_\lambda^a + \frac{1}{4}\epsilon_{abc} A_\rho^a A_\sigma^b A_\lambda^c \right) \right] \right\} \epsilon^{\mu\nu\rho\sigma\lambda}.
\end{aligned}
\tag{9}
$$

Following the conventions of Ref. [9] we shall from now on work with dimensionless quantities, generally denoted by lower case symbols. The relevant definitions (including quantities that will be introduced in the subsequent sections) are collected in table 1[1]. In this table we have abbreviated

$$\lambda_0 \equiv \frac{\lambda}{4\pi}. \tag{10}$$

In particular, in these conventions the dimensionless location of the tip of the cigar-shaped $u$-$x_4$ subspace is

$$u_{\rm KK} = \frac{4}{9}. \tag{11}$$

---

[1]The dimensionless chemical potentials $\mu_B$ and $\mu_I$, which we will refer to as baryon and isospin chemical potentials, are strictly speaking chemical potentials on the quark level, i.e., in addition to the factors given in table 1 they require a factor $N_c$ to be translated to the actual chemical potentials on the baryon level. This is different for the corresponding densities $n_B$ and $n_I$ where the baryonic quantities are obtained by the factors in table 1 without additional factors. This slight inconsistency is retained in order to be consistent with conventions in the previous literature and to keep the notation and terminology as simple as possible. We also note that in Ref. [9], whose notation we otherwise follow closely, the dimensionless baryon density was denoted by $n_I$, where $I$ stands for instanton, whereas in the present work the subscript $I$ is reserved for isospin.

Table 1: Factors that relate the dimensionless quantities in the first row to their dimensionful counterparts, for instance $u = U/[R(M_{KK}R)^2]$, $x_4 = X_4 M_{KK}$ etc.

| $u, u_{KK}, u_T$ $z, \hat{a}_u^{-1}, a_u^{-1}$ | $\hat{a}_0, a_0, \mu_B, \mu_I$ $h, x_0^{-1}$ | $n_B, n_I$ | $x_4, \ell$ | $\hat{a}_i, a_i, x_i^{-1}, t$ | $\Omega$ |
|:---:|:---:|:---:|:---:|:---:|:---:|
| $\dfrac{1}{R(M_{KK}R)^2}$ | $\dfrac{1}{\lambda_0 M_{KK}}$ | $\dfrac{6\pi^2}{N_f \lambda_0^2 M_{KK}^3}$ | $M_{KK}$ | $\dfrac{1}{M_{KK}}$ | $\dfrac{6\pi^2}{N_f N_c \lambda_0^3 M_{KK}^4}$ |

## 2.2 Single baryon with nonzero isospin

Before we introduce our ansatz for baryonic matter it is useful to discuss the case of a single baryon. The simplest way to include baryonic degrees of freedom is to consider D4-branes wrapping the 4-sphere [31]. Due to the presence of the Ramond-Ramond 4-form flux going through this $S^4$ these come with $N_c$ fundamental strings attached, realizing the expected baryon number charge. The other endpoint of these strings is attached to the D8-branes, and by minimizing the energy the baryon vertex gets pulled towards the D8-branes [64]. As a result, these solitonic objects can be seen directly from the point of view of the worldvolume gauge theory as non-abelian instantons [45]. The resulting configurations have been used to extract the spectrum, static properties and form factors of holographic baryons [36, 37]. In particular, in Ref. [36], by quantization of the collective coordinates the baryon spectrum including neutron and proton states was studied. Our many-baryon system will not include this quantization for simplicity, and thus it is useful also in the single-baryon case to only use the semi-classical approximation as a reference for our main results.

For the energy of a single baryon we use the YM approximation of the DBI action and solve the equations of motion (EOMs) in a large-$\lambda$ approximation. This localizes the baryon at the tip of the connected flavor branes, and as a consequence, curvature effects can be neglected, thus yielding the BPST instanton solution with (dimensionless) instanton width $\rho$ and winding number 1, corresponding to baryon number $N_B = 1$. Including a baryon chemical potential $\mu_B$ and an isospin chemical potential $\mu_I$ in the boundary conditions of the temporal components of the gauge fields, the on-shell action yields the dimensionless free energy

$$\phi = \frac{u_{KK}}{3}\left(1 + \frac{\rho^2 \beta}{6u_{KK}^2} + \frac{81u_{KK}}{20\rho^2\lambda_0^2}\right) - \mu_B, \tag{12}$$

from which the dimensionful free energy is obtained by multiplying with $\lambda_0 N_c M_{KK}$, and where we have abbreviated

$$\beta \equiv 1 - \frac{8\lambda_0^2}{3u_{KK}}\mu_I^2. \tag{13}$$

Since the results of this subsection are already contained in Ref. [36] (or can easily be extracted from it), we have skipped all details of the derivation of Eq. (12). In appendix A we do present the detailed derivation for the case of the deconfined geometry, which works analogously and leads to a very similar, but temperature-dependent, result, see Eq. (130).

We see in Eq. (12) that the baryon chemical potential enters in a trivial way. One can read this term as $-\mu_B N_B$ with baryon number $N_B = 1$, which is fixed by construction in this calculation. On the other hand, the isospin chemical potential enters in a more complicated way. To interpret the isospin content we first determine the instanton width by minimizing $\phi$,

$$\rho^2 = \frac{9\sqrt{3}u_{KK}^{3/2}}{\sqrt{10}\lambda_0\beta^{1/2}}. \tag{14}$$

At this minimum, the isospin number is

$$N_I = -\frac{\partial \phi}{\partial \mu_I} = \frac{8N_c \lambda_0 \mu_I}{\sqrt{30} u_{KK}^{1/2} \beta^{1/2}}. \tag{15}$$

This result can now be used to compute the dimensionless baryon energy $e = \phi + \mu_I N_I + \mu_B N_B$ as a function of $N_I$,

$$e = \frac{u_{KK}}{3} + \frac{3u_{KK}^{1/2}}{2\lambda_0} \sqrt{\frac{N_I^2}{6} + \frac{2}{15}}. \tag{16}$$

This result coincides exactly with the first two terms of the mass formula obtained in Ref. [36] by quantizing the instanton configuration, see Eq. (5.26) in that paper, with the quantum number $l + 1$ replaced by the continuous isospin number $N_I N_c$. The additional terms in that equation come from the zero-point energy and the excitations associated with the instanton width and location.

Even though our homogeneous ansatz for isospin-asymmetric baryonic matter will not be based on the instanton solution, Eq. (16) is a very useful reference. It shows, firstly, that the spectrum in the given approximation is continuous in the isospin number and that the lightest state has $N_I = 0$. Therefore, an isospin asymmetry is created continuously even in the single-baryon case, and in the many-baryon case we can expect the system to continuously populate states with nonzero isospin number, in contrast to ordinary nuclear matter, where an isospin asymmetry is achieved by rearranging the population of proton and neutron states.

Secondly, it is instructive to introduce the symmetry energy at this point, which we will then later compute for our holographic baryonic matter. The symmetry energy $S$ is defined as the quadratic term in the expansion of the energy per baryon in the isospin parameter $\delta \equiv N_I/N_B$,

$$\frac{E}{N_B} = \left.\frac{E}{N_B}\right|_{\delta=0} + S\delta^2, \tag{17}$$

or, equivalently,

$$S = \lambda_0 N_c M_{KK} \frac{n_B}{2} \left.\frac{\partial \mu_I}{\partial n_I}\right|_{n_I=0}, \tag{18}$$

with baryon and isospin densities $n_B$ and $n_I$. Using the definition (17) and Eq. (16) together with $E = \lambda_0 N_c M_{KK} e$ and $N_B = 1$ we read off

$$\frac{S}{M_{KK}} = \frac{\sqrt{15}}{8\sqrt{2}} u_{KK}^{1/2} N_c \simeq 0.2282 N_c. \tag{19}$$

Not surprisingly, the symmetry energy of the single-baryon system is of the order of the baryon mass. We shall see later that this remains true for dense baryonic matter in the present approximation.

## 2.3 Homogeneous ansatz for baryonic matter

We now turn to our main goal, the construction of isospin-asymmetric baryonic matter. As introduced in table 1 we denote the dimensionless abelian and non-abelian gauge field components by $\hat{a}_\mu$ and $a_\mu$, with $\mu = 0, 1, 2, 3, u$. Following Refs. [9, 50], we employ the gauge choice $\hat{a}_u = a_u = 0$ and work with the homogeneous, i.e., $\boldsymbol{x}$-independent, ansatz[2]

$$a_i(u) = \frac{\lambda_0 h(u)}{2} \sigma_i, \tag{20}$$

---

[2]The isospin chemical potential we will introduce later breaks the $SU(2)$ symmetry. In the instanton solution this translates into a preferred direction in position space, as realized within the Skyrme model in Ref. [65]. Therefore, one might consider anisotropic configurations with only azimuthal symmetry in the current approximation. Here we will ignore this possibility for simplicity.

where the function $h(u)$ vanishes at the UV boundary $u = \infty$ and has to be determined dynamically. Within this ansatz all gauge fields are functions only of the holographic coordinate $u$. Besides the spatial components of the non-abelian part of the gauge fields (20) also the temporal components $\hat{a}_0(u)$ and $a_0(u)$ are nonvanishing. In particular, and in contrast to Refs. [9, 50], the non-abelian part $a_0(u)$ plays a crucial role since its boundary value encodes the isospin chemical potential. With Eq. (8) we thus arrive at the following nonzero field strengths,

$$\hat{F}_{u0} = i\hat{a}_0', \quad F_{u0}^a = ia_0^{a\prime}, \quad F_{i0}^a = -i\varepsilon_{iab}\lambda_0 h a_0^b, \quad F_{ij}^a = -\varepsilon_{ija}\frac{\lambda_0^2 h^2}{2}, \quad F_{iu}^a = \delta_{ia}\frac{\lambda_0 h'}{2}, \quad (21)$$

where the prime denotes derivative with respect to $u$, and where we have replaced $\hat{a}_0 \to i\hat{a}_0$, $a_0 \to ia_0$ since we work in Euclidean spacetime. (In a slight abuse of notation, we keep using upper case letters for the dimensionless field strengths.)

It is useful to define a new (dimensionless) coordinate $z$ through

$$u^3 = u_{\text{KK}}^3 + u_{\text{KK}} z^2, \quad (22)$$

such that $z \in [-\infty, \infty]$ runs from the UV boundary of the D8-branes to that of the $\overline{\text{D8}}$-branes, with $z = 0$ corresponding to the tip of the connected branes at $u = u_{\text{KK}}$. In the following we shall switch between the two variables depending on which one is more convenient for a particular calculation or argument. For example, most derivations are more compactly written in the $u$ variable, while for some properties of our solutions, such as their symmetry across the two halves of the flavor branes, it is unavoidable to employ the $z$ parametrization.

Using the definition based on the topological winding number, the baryon number density is

$$\frac{N_B}{V} = -\frac{M_{\text{KK}}^3}{8\pi^2} \int_{-\infty}^{\infty} dz \, \text{Tr}[F_{ij}F_{kz}]\epsilon_{ijk} = \frac{\lambda_0^3 M_{\text{KK}}^3}{8\pi^2} \int_{-\infty}^{\infty} dz \, \partial_z(h^3), \quad (23)$$

where $V$ is the three-volume. We see that within our simple ansatz no net baryon number is generated unless $h(z)$ is discontinuous. (This discontinuity can be avoided by introducing an ansatz on the level of the field strengths, not the gauge fields [51].) We introduce the discontinuity at the tip of the connected branes, $z = 0$, and require $h(z)$ to be antisymmetric under $z \to -z$, with boundary conditions

$$h(z = 0^\pm) = \pm h_c, \qquad h(z = \pm\infty) = 0. \quad (24)$$

If baryonic matter is described with the help of instantons, it was shown that different layers appear as the density is increased [10, 12, 48, 51]. In our current apporach this might be included by introducing more than one discontinuity with dynamically determined locations in the holographic direction. Here we only consider a single discontinuity for simplicity.

For the practical calculation, we can restrict ourselves to one half of the connected branes, say $z \geq 0$ and work with the function $h(u)$ with boundary conditions $h(u_{\text{KK}}) = h_c$ and $h(\infty) = 0$. With the help of Eq. (23) we can relate the IR boundary condition $h_c$ to the baryon density,

$$n_B = -\frac{3}{4}\lambda_0 h_c^3, \quad (25)$$

with the dimensionless baryon density $n_B$ from table 1. We see that for positive baryon densities $h_c < 0$.

We now insert our ansatz into the action (3), use the YM approximation for the DBI action (4) and notice that only the first term of the CS action (9) with the structure $\hat{A}FF$ contributes. Omitting the term constant in the fields this yields

$$S = \mathcal{N}N_f \frac{V}{T} \int_{u_{\text{KK}}}^{\infty} du \, \mathcal{L}, \quad (26)$$

with

$$\mathcal{N} = \frac{N_c M_{KK}^4 \lambda_0^3}{6\pi^2} \,, \tag{27}$$

and the Lagrangian

$$\mathcal{L} = \frac{u^{5/2}}{2\sqrt{f}} \left( g_1 - f\hat{a}_0'^2 - f a_0'^2 + g_2 - g_3 \right) - \frac{9}{4}\lambda_0 \hat{a}_0 h^2 h' \,, \tag{28}$$

where we have abbreviated[3]

$$g_1 \equiv \frac{3f h'^2}{4} \,, \qquad g_2 \equiv \frac{3\lambda_0^2 h^4}{4u^3} \,, \qquad g_3 \equiv \frac{2\lambda_0^2 h^2 a_0^2}{u^3} \,, \tag{29}$$

with $a_0^2 = a_0^a a_0^a$ and $a_0'^2 = a_0^{a\prime} a_0^{a\prime}$. We shall introduce the isospin chemical potential in the $\sigma^3$ direction, such that it is consistent to set the $\sigma^1$ and $\sigma^2$ components of $a_0$ to zero, and we denote $a_0 \equiv a_0^3$ from now on.

From the action (26) we derive the following EOMs for $\hat{a}_0$, $a_0$ and $h$,

$$\partial_u \left( u^{5/2} \sqrt{f} \hat{a}_0' \right) = \frac{9}{4}\lambda_0 h^2 h' \,, \tag{30a}$$

$$\partial_u \left( u^{5/2} \sqrt{f} a_0' \right) = \frac{2\lambda_0^2 h^2 a_0}{u^{1/2}\sqrt{f}} \,, \tag{30b}$$

$$\frac{3}{2}\partial_u \left( u^{5/2} \sqrt{f} h' \right) - \frac{9\lambda_0 h^2 n_B Q}{2u^{5/2}\sqrt{f}} = \frac{\lambda_0^2 h}{u^{1/2}\sqrt{f}}(3h^2 - 4a_0'^2) \,, \tag{30c}$$

where we have defined

$$Q(u) \equiv 1 - \frac{h^3(u)}{h_c^3} \,. \tag{31}$$

The EOM for the abelian gauge field (30a) can easily be integrated to obtain

$$\hat{a}_0' = \frac{n_B Q}{u^{5/2}\sqrt{f}} \,, \tag{32}$$

where the integration constant is the baryon density (25). The other two equations of motion, which couple $a_0$ and $h$, need to be solved numerically.

The thermodynamic potential (= free energy density) is then obtained from the on-shell action. We define the dimensionless thermodynamic potential by

$$\Omega = \int_{u_{KK}}^{\infty} du\, \mathcal{L} = \frac{1}{2}\int_{-\infty}^{\infty} dz\, \frac{\partial u}{\partial z}\mathcal{L} \,, \tag{33}$$

where

$$\frac{\partial u}{\partial z} = \frac{2u_{KK}|z|}{3u^2} \,, \tag{34}$$

and where $\mathcal{L}$ is evaluated at the stationary point. The dimensionful free energy density is then obtained by multiplication with $\mathcal{N}N_f$. In the present YM approximation, $\Omega$ is finite and does not require a vacuum subtraction. In the vacuum, where $h = \hat{a}_0' = a_0' = 0$ we have $\Omega = 0$.

---

[3]This notation was also used in Ref. [9], and for $g_3 = 0$ we recover the action in this reference. However, within the definitions of $g_1$ and $g_2$ we differ by a factor $N_f = 2$ from Ref. [9] because in that reference the flavor trace was taken within the square root only over the non-abelian terms.

## 2.4 Including pion condensation

The chemical potentials of the boundary field theory are introduced through the UV boundary conditions for the temporal components of the gauge field. Since the D8-$\overline{\text{D8}}$ pairs join in the bulk there is only a single $U(2)$ gauge field. However, in the UV chiral symmetry is effectively restored, so that the gauge field is allowed to behave differently for left-handed fermions at $z \to +\infty$ and right-handed fermions at $z \to -\infty$. Let us denote the $U(2)$-valued boundary conditions by

$$\mu_{L,R} = \hat{a}_0(\pm\infty)\mathbb{1} + a_0(\pm\infty)\sigma^3 \,. \tag{35}$$

In order to implement a baryon chemical potential we set $\hat{a}_0(\pm\infty) = \mu_B$. In contrast, for the isospin chemical potential it will be necessary to consider the possibility of having either equal or opposite boundary conditions at the left- and right-handed boundaries. Let us briefly review the arguments of Ref. [52] to explain this. As it was shown in Ref. [31], one recovers the chiral Lagrangian for massless pions from the Witten-Sakai-Sugimoto model by expanding in radial modes and carrying out the integral in $z$,

$$\mathcal{L}_{\text{chiral}} = \frac{f_\pi^2}{4} \text{Tr} \left[ D_\mu \Sigma D^\mu \Sigma^\dagger \right] \,, \tag{36}$$

where the pion decay constant is given in terms of the parameters of the Witten-Sakai-Sugimoto model by

$$f_\pi^2 = \frac{\lambda M_{\text{KK}}^2 N_c}{54\pi^4} \,, \tag{37}$$

and the pion matrix $\Sigma$ can be expressed as the holonomy

$$\Sigma = \mathcal{P} \exp \left( i \int_{-\infty}^{+\infty} dz \, a_z \right) \,, \tag{38}$$

where $\mathcal{P}$ denotes path ordering. The chemical potentials are introduced through the covariant derivative in Eq. (36). Since pions do not carry baryon number, $\mu_B$ simply drops out of this Lagrangian and thus, if only $\mu_B$ is nonzero, the vacuum is $\Sigma = \mathbb{1}$. As Eq. (38) shows, this is consistent with our gauge choice $a_z = 0$. An isospin chemical potential $\mu_I$, however, induces an effective potential for the pions through the covariant derivative $D_\nu\Sigma = \partial_\nu\Sigma - i\mu_I\delta_{\nu 0}[\sigma^3, \Sigma]$, resulting in a nontrivial minimum $\Sigma = \Sigma_{\text{min}} \propto \sigma^{1,2}$. This minimum corresponds to a condensate of charged pions (which, since here $m_\pi = 0$, already occurs at infinitesimally small $\mu_I$). It seems this is in conflict with our gauge choice $a_z = 0$. Fortunately, we may employ a global chiral transformation $\Sigma \to g_L^{-1}\Sigma g_R$ where $g_L \in U(2)_L$ and $g_R \in U(2)_R$, which leaves the potential invariant, to work in a frame where the transformed minimum is trivial, $g_L^{-1}\Sigma_{\text{min}}g_R = \mathbb{1}$. For example, one can choose $g_L = \Sigma_{\text{min}}$ and $g_R = \mathbb{1}$. This transformation affects the left- and right-handed chemical potentials, which transform as $\mu_L \to g_L^{-1}\mu_L g_L$ and $\mu_R \to g_R^{-1}\mu_R g_R$. As a consequence, one finds that a *vector* isospin chemical potential $\mu_L = \mu_R = \mu_I\sigma^3$ in the original frame is seen after our transformation as *axial*, $-\mu_L = \mu_R = \mu_I\sigma^3$. Thus, in order to study pion condensation in the presence of a vector isospin chemical potential, we may keep the $a_z = 0$ gauge but have to impose axial boundary conditions for the isospin component. Vector boundary conditions in the isospin component correspond to a chirally broken phase without pion condensation. In other words, rather than keeping the boundary conditions fixed and vary the chiral field we fix the chiral field and vary the boundary conditions according to the transformation that is needed to keep the chiral field fixed. We collect the boundary conditions for all relevant functions in table 2, where, following the terminology of Ref. [53], we refer to the two types of boundary conditions as $\sigma$ and $\pi$. Baryonic matter can be added in both cases, i.e., with and without pion condensation, via the function $h(z)$, and in each case we require the boundary condition (24), which is also included in the table. For completeness, the table

Table 2: Boundary conditions at $z = \pm\infty$ for the various components of the gauge fields and the embedding function of the flavor branes $x_4$ (the latter is only relevant for the deconfined geometry discussed in Sec. 4). Boundary conditions of type $\pi$ ($\sigma$) do (do not) include pion condensation.

| type | $\hat{a}_0(\pm\infty)$ | $a_0(\pm\infty)$ | $h(\pm\infty)$ | $x_4(\pm\infty)$ |
|---|---|---|---|---|
| $\sigma$ | $\mu_B$ | $\mu_I$ | 0 | $\pm\ell/2$ |
| $\pi$ | $\mu_B$ | $\pm\mu_I$ | 0 | $\pm\ell/2$ |

also gives the boundary condition for the embedding function of the flavor branes, which is irrelevant in the present section due to the antipodal separation of the flavor branes but which will become relevant when we discuss the deconfined geometry in Sec. 4.

## 2.5 Free energy density

Next, we derive an expression for the free energy density and show that the usual thermodynamic relations with respect to baryon and isospin number densities are respected in our approximation. We also derive expressions for the isospin density and the baryon chemical potential which are useful for the practical evaluation. To this end, we need to discuss the IR behavior of the functions $\hat{a}_0$, $a_0$ and $h$. The series expansions about $u = u_{\rm KK}$ (and thus $z = 0$) of $a_0$ and $h$ can be written as

$$a_0(u) = a_c + a_{(1)}\sqrt{u - u_{\rm KK}} + a_{(2)}(u - u_{\rm KK}) + \ldots = a_c + \frac{a_{(1)}}{\sqrt{3u_{\rm KK}}}z + \frac{a_{(2)}}{3u_{\rm KK}}z^2 + \ldots, \quad (39a)$$

$$h(u) = h_c + h_{(1)}\sqrt{u - u_{\rm KK}} + h_{(2)}(u - u_{\rm KK}) + \ldots = h_c + \frac{h_{(1)}}{\sqrt{3u_{\rm KK}}}z + \frac{h_{(2)}}{3u_{\rm KK}}z^2 + \ldots. \quad (39b)$$

The EOMs (30b) and (30c) can be used to express all higher order coefficients $a_{(2)}, a_{(3)}, \ldots$ and $h_{(2)}, h_{(3)}, \ldots$ recursively in terms of $a_c, a_{(1)}, h_c, h_{(1)}$. From Eq. (30a) we obtain the expansion for the abelian component $\hat{a}_0$,

$$\hat{a}_0(u) = \hat{a}_c + \hat{a}_{(2)}(u - u_{\rm KK}) + \hat{a}_{(3)}(u - u_{\rm KK})^{3/2} + \ldots = \hat{a}_c + \frac{\hat{a}_{(2)}}{3u_{\rm KK}}z^2 + \frac{\hat{a}_{(3)}}{(3u_{\rm KK})^{3/2}}z^3 \ldots, \quad (40)$$

where $\hat{a}_{(2)}$ and $\hat{a}_{(3)}$ can be written in terms of the coefficients of the series expansion of $h$,

$$\hat{a}_{(2)} = \frac{3\sqrt{3}\lambda_0 h_c^2 h_{(1)}}{4u_{\rm KK}}, \qquad \hat{a}_{(3)} = \frac{\sqrt{3}\lambda_0 h_c(h_{(1)}^2 + h_c h_{(2)})}{2u_{\rm KK}}. \quad (41)$$

All expressions are valid on the $z > 0$ half of the connected flavor branes. We can extend them over both halves as follows. The discontinuity in $h$ is implemented by using $-h(|z|)$ for the $z < 0$ half, where $h(z)$ is the solution on the $z > 0$ half. The resulting function is thus odd in $z$. Its IR boundary value $\pm h_c$ is given by the baryon density, see Eq. (25), and $h_{(1)}$ must be determined from the numerical solution of the EOMs. In both types of boundary conditions we consider, $\hat{a}_0(z)$ is even in $z$. [Note that changing the sign of $h$ on the $z < 0$ half results in a sign flip of the coefficient $\hat{a}_{(3)}$, but not of $\hat{a}_{(2)}$, leading to the correct parity of the expansion (40).] Hence, $\hat{a}_0(z)$ is automatically smooth at $z = 0$ since there is no linear term in the expansion (40). The boundary value $\hat{a}_c$ has to be determined dynamically. Finally, the parity of $a_0$ depends on the type of boundary conditions. For $\sigma$-type conditions we require $a_0$ to be even in $z$. In this case, $a_c$ is determined dynamically, and we will show below that minimizing the free energy with respect to $a_c$ yields $a_{(1)} = 0$. Hence also $a_0$ turns out to be smooth at

$z = 0$. For $\pi$-type boundary conditions we require $a_0$ to be odd in $z$. Now $a_{(1)}$ will adjust itself to a nonzero value according to the EOMs and we will see that we need to set $a_c = 0$. Thus $a_0$ is continuous and smooth at $z = 0$ also in this case.

In order to verify the usual thermodynamic relations and to minimize the free energy with respect to the parameters $h_c$ and $a_c$, we compute the derivative $\Omega$ with respect to $x = \mu_B, \mu_I, h_c, a_c$. With the help of the EOMs we obtain

$$
\begin{aligned}
\frac{\partial \Omega}{\partial x} &= \frac{1}{2} \int_{-\infty}^{\infty} dz \, \partial_z \left( \frac{\partial \mathcal{L}}{\partial \hat{a}_0'} \frac{\partial \hat{a}_0}{\partial x} + \frac{\partial \mathcal{L}}{\partial a_0'} \frac{\partial a_0}{\partial x} + \frac{\partial \mathcal{L}}{\partial h'} \frac{\partial h}{\partial x} \right) \\
&= \frac{1}{2} \int_{-\infty}^{\infty} dz \left[ -\partial_z \left( u^{5/2} \sqrt{f} \, \hat{a}_0' \frac{\partial \hat{a}_0}{\partial x} \right) - \partial_z \left( u^{5/2} \sqrt{f} \, a_0' \frac{\partial a_0}{\partial x} \right) \right. \\
&\qquad \left. + \partial_z \left( \frac{3u^{5/2}\sqrt{f}h' - 9\lambda_0 \hat{a}_0 h^2}{4} \frac{\partial h}{\partial x} \right) \right],
\end{aligned}
\tag{42}
$$

where, although the integration variable is $z$, the functions are written in terms of $u$ for compactness (throughout the paper prime stands for the derivative with respect to $u$). The integral gives rise not only to $z = \pm\infty$ contributions but also to terms coming from $z = 0$ since $h$ is discontinuous there. For the first term we use that for all phases we consider $\hat{a}_0(z = \pm\infty) = \mu_B$, and that $\hat{a}_0'$ is odd in $z$ because $\hat{a}_0$ is even (and because $\partial_u z$ (34) is even). Therefore, using Eq. (31), we have $u^{5/2}\sqrt{f}\hat{a}_0' = \pm n_B$ for $z = \pm\infty$. For the second term we recall that $a_0(z = \pm\infty) = \mu_I$ for $\sigma$-type boundary conditions and $a_0(z = \pm\infty) = \pm\mu_I$ for $\pi$-type boundary conditions. However, this difference in sign is canceled by $a_0'$, which has opposite parity (odd for $\sigma$-type and even for $\pi$-type). This term thus gives nonzero contributions from the UV boundaries and from $z = 0$. Finally, the only contribution to the third term comes from the discontinuity at $z = 0$ because the boundary terms at $z = \pm\infty$ vanish. We use that $h'$ is even in $z$ and $h(z \to 0^\pm) = \pm h_c$. Putting all this together, we find

$$
\begin{aligned}
\frac{\partial \Omega}{\partial x} &= -n_B \frac{\partial \mu_B}{\partial x} - \left( u^{5/2} \sqrt{f} a_0' \right)_{u=\infty} \frac{\partial \mu_I}{\partial x} + \frac{\sqrt{3}u_{KK}^2 a_{(1)}}{2} \frac{\partial a_c}{\partial x} \\
&\quad - \left( \frac{3\sqrt{3}u_{KK}^2 h_{(1)}}{8} - \frac{9\lambda_0 \hat{a}_c h_c^2}{4} \right) \frac{\partial h_c}{\partial x} .
\end{aligned}
\tag{43}
$$

This result only requires information from one half of the connected branes, so that we can go back to working in the $u$ coordinate (on the $z > 0$ half of the branes).

We expect $x = \mu_B, \mu_I$ to yield the thermodynamic relations

$$
\frac{\partial \Omega}{\partial \mu_B} = -n_B, \qquad \frac{\partial \Omega}{\partial \mu_I} = -n_I .
\tag{44}
$$

Setting $x = \mu_B$ in Eq. (43) is simply a consistency check and gives no additional information. The second relation defines $n_I$, the dimensionless isospin density. It is related to its dimensionful counterpart by the same factor as for $n_B$, see table 1, as can be seen by inserting the dimensionful factors for $\Omega$ and $\mu_B, \mu_I$ into Eq. (44). We find

$$
n_I = \left( u^{5/2} \sqrt{f} a_0' \right)_{u=\infty} = \frac{\sqrt{3}}{2} u_{KK}^2 a_{(1)} + 2\lambda_0^2 \int_{u_{KK}}^{\infty} du \, \frac{h^2 a_0}{u^{1/2}\sqrt{f}},
\tag{45}
$$

where we have made use of Eq. (30b). We see that the thermodynamic relations are consistent with the AdS/CFT dictionary: both $n_B$ and $n_I$ are given by the subleading terms at the

holographic boundary,

$$\hat{a}_0' = \frac{n_B}{u^{5/2}} + \dots, \qquad a_0' = \frac{n_I}{u^{5/2}} + \dots. \tag{46}$$

To minimize the free energy with respect to $h_c$ we set $x = h_c$ in Eq. (43) and obtain

$$\hat{a}_c = \frac{u_{KK}^2 h_{(1)}}{2\sqrt{3}\lambda_0 h_c^2}. \tag{47}$$

This result allows us to write $\mu_B$ as

$$\mu_B = \hat{a}_c + \int_{u_{KK}}^{\infty} du\, \hat{a}_0' = \frac{u_{KK}^2 h_{(1)}}{2\sqrt{3}\lambda_0 h_c^2} + \int_{u_{KK}}^{\infty} du\, \frac{n_B Q}{u^{5/2}\sqrt{f}}, \tag{48}$$

where Eq. (32) has been used. Finally, we can set $x = a_c$. Again, we expect the derivative to vanish in this case. For the $\sigma$-type boundary conditions, where $a_c$ adjust itself dynamically, we find $a_{(1)} = 0$, which is the smoothness condition for $a_0$. For the $\pi$-type boundary condition in the UV we must require $a_c = 0$ as an additional IR boundary condition to begin with, which implies continuity and smoothness for $a_0$, and in this case $a_{(1)}$ can only be computed numerically. Applying this conclusion to the isospin density (45), we see that in the absence of pion condensation, where $a_{(1)} = 0$, the only contribution comes from the integral, which only is nonzero for a nonzero function $h(u)$, i.e., in the presence of baryons. In the pion-condensed phase, however, where $a_{(1)}$ is nonzero, isospin density is also generated in the absence of baryons, as it should be.

We can use these relations to compute an explicit form of the free energy (33). With the help of partial integration and the EOMs we find

$$\Omega = \int_{u_{KK}}^{\infty} du\, \frac{u^{5/2}}{2\sqrt{f}} \left[ g_1 + g_2 + \frac{(n_B Q)^2}{u^5} \right] - \mu_B n_B - \frac{\mu_I n_I}{2}. \tag{49}$$

This is a useful compact form to compute $\Omega$ numerically. The factor $1/2$ in the last term has no particular meaning, extracting an additional $-\mu_I n_I/2$ from the integral is possible, but would result in a more complicated integrand.

## 2.6 Possible phases

In the previous subsection we have kept the notation general such that Eqs. (45), (48) and (49) are valid for all phases we consider, in particular for both types of boundary conditions explained in Sec. 2.4. We now describe all distinct phases included in this analysis.

- **Vacuum:** The vacuum configuration is defined by vanishing baryon and isospin densities, $n_B = n_I = 0$. The boundary conditions are of the $\sigma$ type, and the solutions to the EOMs are simply constants,

$$h(u) = 0, \qquad \hat{a}_0(u) = \mu_B, \qquad a_0(u) = \mu_I. \tag{50}$$

  In this phase, the free energy density is zero, $\Omega = 0$.

- **Pion-condensed phase ($\pi$):** Here we have $n_B = 0$ and a nonzero isospin density, which is created by a pion condensate. This phase requires $\pi$-type boundary conditions, and the solutions of the EOMs are

$$h(u) = 0, \qquad \hat{a}_0(u) = \mu_B, \qquad a_0(u) = \frac{2\mu_I}{\pi} \arctan\sqrt{\frac{u^3}{u_{KK}^3} - 1}. \tag{51}$$

To compute the isospin density we can simply expand $a_0(u)$ about $u_{\rm KK}$ to find the coefficient $a_{(1)}$ and insert the result into Eq. (45). The free energy density is computed from Eq. (49) and we obtain

$$n_I = \frac{3u_{\rm KK}^{3/2}}{\pi}\mu_I\,, \qquad \Omega = -\frac{3u_{\rm KK}^{3/2}}{2\pi}\mu_I^2\,. \tag{52}$$

As a check, one can use these expressions to confirm the thermodynamic relation (44) for the isospin density. We also see that the pion-condensed phase is preferred over the vacuum for any $|\mu_I| > 0$, as expected in the chiral limit. With the expression for $f_\pi$ (37) the relation for the isospin density in Eq. (52) implies

$$\left(\frac{N_c N_f \lambda_0^2 M_{\rm KK}^3}{6\pi^2}n_I\right) = 4f_\pi^2\left(\lambda_0 M_{\rm KK}\mu_I\right)\,, \tag{53}$$

where the expressions in parentheses are the physical dimensionful quantities defined through table 1. This form of the isospin density is in agreement with chiral perturbation theory [21][4].

- **Pure baryonic phase (B):** Here we work with $\sigma$-type boundary conditions, i.e., pions do not condense. Isospin number comes solely from baryonic matter and is induced by the non-trivial profile of $a_0$, which couples to $h$ through the EOMs (30). The B phase is a direct generalization of the baryonic phase studied in Refs. [9,50] to nonzero isospin. Here, the solutions of the EOMs and the value of the free energy have to be computed numerically. Exemplary profiles that illustrate the shape and symmetry of the solutions are shown in the left panel of Fig. 1. We discuss the results more systematically in Sec. 3 and briefly explain the numerical procedure for solving the EOMs at the beginning of that section.

- **Coexistence phase ($\pi$B):** Also in this case both number densities are allowed to be nonzero, this time with $\pi$-type boundary conditions, such that $a_0(z)$ is antisymmetric. In this phase baryonic matter coexists with a pion condensate, and both contribute to the isospin density. Therefore, $n_I$ remains finite in the $n_B \to 0$ limit, thus reproducing the $\pi$-phase above. The evaluation of this phase also has to be done numerically. Since the only difference to the pure baryonic phase are the boundary conditions, the numerical calculation is very similar. The profiles of the gauge fields for a particular parameter set are shown in the right panel of Fig. 1.

Both isospin-asymmetric baryonic phases represent novel configurations in the Witten-Sakai-Sugimoto model. For any given $\mu_B$ and $\mu_I$ we may now calculate their thermodynamic properties and determine the energetically preferred phase. The results will be discussed in Sec. 3.

Dense matter in a compact star lives on a curve in the $\mu_B$-$\mu_I$ plane because of the constraints of beta equilibrium – which relates $\mu_I$ to the electron chemical potential – and electric charge neutrality – which fixes the electron chemical potential for any given $\mu_B$. As we have discussed in Sec. 2.2, our present holographic approach cannot be interpreted as a system of neutrons and protons since this would require the quantization of instanton solutions in the bulk. Nevertheless, it is illustrative to assume that our two isospin components correspond to neutron and proton states simply by assigning electric charges 0 and +1 to them. This will give

---

[4]For this comparison it is important to note that in our convention $\mu_B = (\mu_u + \mu_d)/2$ and $\mu_I = (\mu_u - \mu_d)/2$, with the quark chemical potentials $\mu_u = \hat{a}_0(\infty) + a_0(\infty)$, $\mu_d = \hat{a}_0(\infty) - a_0(\infty)$. In this convention, the zero-temperature onset of pion condensation in the presence of a pion mass occurs at $m_\pi/2$.

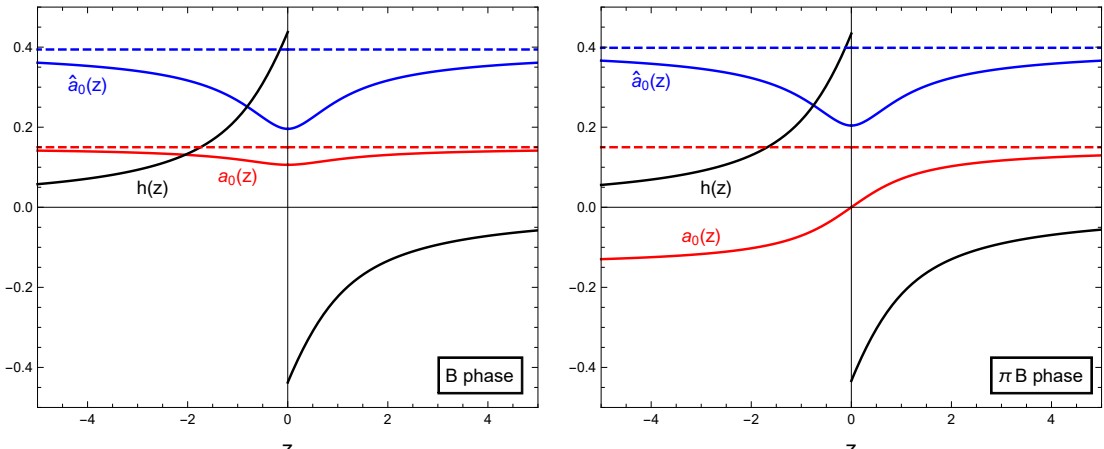

Figure 1: Profiles of the functions $h(z)$ (black, non-abelian spatial component of the gauge field), $\hat{a}_0(z)$ (blue, abelian temporal gauge field) and $a_0(z)$ (red, non-abelian temporal gauge field). The chemical potentials act as boundary conditions, here chosen to be $\mu_B = 0.4$ and $\mu_I = 0.15$, indicated by the dashed lines. The discontinuity in $h(z)$ gives rise to a nonzero baryon number. In the left panel $a_0(z)$ is even in $z$ ($\sigma$-type boundary conditions, pure baryon phase B), while in the right panel $a_0(z)$ is odd in $z$ ($\pi$-type boundary conditions, coexistence phase $\pi$B). The resulting baryon and isospin densities are $(n_B, n_I) \simeq (0.08, 0.02)$ and $(n_B, n_I) \simeq (0.07, 0.05)$ respectively. In both panels we have used $\lambda = 15$.

us an idea of how compact star conditions affect our solutions and may be useful as a reference for future studies that include neutron and proton states in a more realistic way. To this end, we restrict ourselves to the B phase. The reason is that in the $\pi$B phase we cannot easily separate baryon from pion contributions to assign different electric charges to them. Moreover, although pion condensation in nuclear matter was already envisioned a long time ago [66,67], it remains unclear whether the conditions in dense neutron star matter are favorable for pions to condense, see for instance Ref. [68].

Equilibrium in ordinary nuclear matter with respect to beta decay and electron capture relates the electron chemical potential $\mu_e$ to the neutron and proton chemical potentials, $\mu_e = \mu_n - \mu_p$, and the electron chemical to the muon chemical potential $\mu_e = \mu_\mu$, where we have neglected the neutrino chemical potential, which is a good approximation except for the very early stages in the life of the star. The lepton chemical potentials give rise to the corresponding electron and muon number densities,

$$n_e = \frac{\mu_e^3}{3\pi^2}, \qquad n_\mu = \Theta(\mu_e - m_\mu)\frac{(\mu_e^2 - m_\mu^2)^{3/2}}{3\pi^2}, \tag{54}$$

where we have neglected the electron mass, and $m_\mu \simeq 106\,\text{MeV}$ is the muon mass. The charge neutrality condition is then $n_p = n_e + n_\mu$, where $n_p$ is the proton number density. We now identify the difference between neutron and proton chemical potentials (divided by 2) with our isospin chemical potential $\mu_I$, such that, using table 1 to turn $\mu_I$ into its dimensionful version, beta equilibrium reads

$$\mu_I = \frac{\mu_e}{2\lambda_0 N_c M_{\text{KK}}}. \tag{55}$$

Then, identifying the proton number density with $(n_B - n_I)/2$, the neutrality condition becomes

$$n_B - n_I = \frac{6\pi^2(n_e + n_\mu)}{\lambda_0^2 M_{\text{KK}}^3} = 16 N_c^3 \lambda_0 \mu_I^3 \left[1 + \Theta(\mu_e - m_\mu)\left(1 - \frac{m_\mu^2}{\mu_e^2}\right)^{3/2}\right]. \tag{56}$$

Due to the additional mass scale $m_\mu$, the muon contribution requires us to choose a value for $M_{KK}$, which is not the case if only (approximately massless) electrons are included. For given $n_B$, Eq. (56) can be solved for $\mu_I$ because $n_I$ is a (complicated) function of $n_B$ and $\mu_I$, see Eq. (45). Then, the solution is used to determine the associated $\mu_B$ via Eq. (48).

## 2.7 Low-density approximation

Solving Eqs. (30) requires numerical methods in general if baryons are present. In the limit of small baryon and isospin densities, however, one finds an analytical solution. Even though we shall see that this solution can only be applied in an unstable regime, we will gain some insight from the analytical expressions and may use them as a benchmark for the numerics. We assume $\mu_I$ and thus $a_0(u)$ to be small, say of order $\epsilon$, and assume $h(u)$ to be of the same order. Since $n_B \propto h_c^3$ the baryon density is then of order $\epsilon^3$, while $n_I$ is of order $\epsilon$ in the $\pi$B phase and of order $\epsilon^3$ in the B phase on account of Eq. (45). Then, from Eq. (48) we see that the leading-order behavior of the baryon chemical potential is $\mu_B \propto 1/\epsilon$. This simple power counting argument already shows that the baryon density will *decrease* with increasing baryon chemical potential, which indicates a thermodynamical instability.

Within this approximation, the EOMs (30) become to lowest order in $\epsilon$

$$\partial_u \left( u^{5/2} \sqrt{f} \hat{a}_0' \right) \simeq \partial_u \left( u^{5/2} \sqrt{f} a_0' \right) \simeq \partial_u \left( u^{5/2} \sqrt{f} h' \right) \simeq 0. \tag{57}$$

Thus, all three functions have the form $c_1 + c_2 \arctan\sqrt{u^3/u_{KK}^3 - 1}$, and the only difference between them comes from the boundary conditions, which determine the integration constants $c_1$ and $c_2$. We find

$$\hat{a}_0(u) \simeq \mu_B, \qquad h(u) \simeq -\left( \frac{4n_B}{3\lambda_0} \right)^{1/3} \left( 1 - \frac{2}{\pi} \arctan\sqrt{\frac{u^3}{u_{KK}^3} - 1} \right), \tag{58}$$

and

$$a_0(u) \simeq \begin{cases} \mu_I & \text{B phase} \\ \dfrac{2\mu_I}{\pi} \arctan\sqrt{\dfrac{u^3}{u_{KK}^3} - 1} & \pi\text{B phase} \end{cases}. \tag{59}$$

This yields the leading-order contribution to the baryon chemical potential from Eq. (48),

$$\mu_B \simeq \frac{u_{KK}^{3/2}}{\pi} \left( \frac{3}{4\lambda_0^2 n_B} \right)^{1/3}, \tag{60}$$

and the leading-order contribution to the isospin density from Eq. (45),

$$n_I \simeq \begin{cases} \dfrac{2\alpha u_{KK}^{7/2} \mu_I}{\pi^2 \mu_B^2} & \text{B phase} \\ \dfrac{3 u_{KK}^{3/2} \mu_I}{\pi} & \pi\text{B phase} \end{cases}, \tag{61}$$

where we have abbreviated the numerical factor

$$\alpha \equiv \int_1^\infty \frac{du\, u}{\sqrt{u^3 - 1}} \left( 1 - \frac{2}{\pi} \arctan\sqrt{u^3 - 1} \right)^2 \simeq 0.455359. \tag{62}$$

The dimensionless free energy (49) can be approximated by

$$\Omega \simeq \int_{u_{\mathrm{KK}}}^{\infty} du \frac{u^{5/2} g_1}{2\sqrt{f}} - \mu_B n_B - \frac{\mu_I n_I}{2} \simeq \begin{cases} \dfrac{3 u_{\mathrm{KK}}^{9/2}}{8\pi^3 \lambda_0^2} \dfrac{1}{\mu_B^2} - \dfrac{\alpha u_{\mathrm{KK}}^{7/2}}{\pi^2} \dfrac{\mu_I^2}{\mu_B^2} & \text{B phase} \\[3ex] \dfrac{3 u_{\mathrm{KK}}^{9/2}}{8\pi^3 \lambda_0^2} \dfrac{1}{\mu_B^2} - \dfrac{3 u_{\mathrm{KK}}^{3/2}}{2\pi} \mu_I^2 & \pi\text{B phase} \end{cases} . \quad (63)$$

Here we have kept the leading contribution of order $\epsilon^2$ in both cases, and in the B phase also the $\mu_I$ dependent part of the subleading $\epsilon^4$ contribution, such that the thermodynamic relations (44) are fulfilled at leading order for both baryon and isospin number.

As already anticipated, Eq. (60) confirms that the approximation is only valid in a regime where the baryon number goes to zero as the chemical potential is increased. This is not only an unstable branch of the solution, it also indicates a well-known shortcoming of the present homogeneous ansatz for baryonic matter: One would expect $\mu_B$ to approach the vacuum mass of the baryon as $n_B \to 0$. In other words, in our approach the vacuum mass is infinite. Since the ansatz is expected to work well at large baryon densities it is not surprising that unphysical results can arise in the low-density regime.

We may further exploit our low-density approximation to investigate the symmetry energy. Here we focus on the B phase since the isospin contribution to the $\pi$B free energy in Eq. (63) is a pure pion contribution and thus in this phase we do not learn anything about baryonic matter from computing the symmetry energy in the present approximation. With the dimensionless energy density $\varepsilon = \Omega + \mu_B n_B + \mu_I n_I$ we find for the energy per baryon in the B phase

$$\frac{\varepsilon}{n_B} \simeq \frac{3 u_{\mathrm{KK}}^{3/2}}{2\pi} \left( \frac{3}{4\lambda_0^2 n_B} \right)^{1/3} \left[ 1 + \frac{\pi}{6\alpha u_{\mathrm{KK}}^2} \left( \frac{3 n_B^2}{4\lambda_0^2} \right)^{1/3} \frac{n_I^2}{n_B^2} \right] . \quad (64)$$

This result can be compared to the single baryon energy (16). In particular, we can read off the symmetry energy

$$\frac{S}{M_{\mathrm{KK}}} \simeq \frac{3 N_c}{8\alpha u_{\mathrm{KK}}^{1/2}} \left( \frac{n_B}{6\lambda_0} \right)^{1/3} . \quad (65)$$

We shall compare this low-density expression to the full numerical result in Sec. 3.3. As we shall see, the $n_B^{1/3}$ behavior is a reasonable qualitative indication for the symmetry energy even at larger densities, though the actual result does deviate quantitatively.

We can also use the low-density results for an illustration of how the neutrality condition and the beta equilibrium affect our holographic matter. Neglecting the muon contribution for simplicity, the two conditions (55) and (56) together with the low density expressions (60) and (61) yield

$$1 = \frac{3p}{2^{1/3}} x_P^{1/3} + 2 x_P , \quad (66)$$

where we have abbreviated

$$p \equiv \alpha u_{\mathrm{KK}}^{1/2} \left( \frac{2}{3} \right)^{5/3} \frac{\lambda_0}{N_c} , \quad (67)$$

and where

$$x_P \equiv \frac{n_B - n_I}{2 n_B} = \frac{\mu_e^3}{\lambda_0^2 n_B M_{\mathrm{KK}}^3} , \quad (68)$$

is the "proton" fraction (more precisely, since our system does not have proton states, the fraction of baryonic matter in the isospin component that we have assumed to behave like a

proton in terms of electric charge and beta decay). We can solve Eq. (66) to find

$$x_P = \frac{[p-(\sqrt{1+p^3}-1)^{2/3}]^3}{4(\sqrt{1+p^3}-1)} = \begin{cases} \dfrac{1}{2} - \dfrac{3p}{2^{5/3}} + \dots & \text{for } p \to 0 \\[2mm] \dfrac{2}{27p^3} + \dots & \text{for } p \to \infty \end{cases}. \tag{69}$$

We see that for small $\lambda/N_c$ we approach symmetric nuclear matter. This suggests that in this case the symmetry energy is very large, the system prefers to have the same numbers of protons and neutrons despite the conditions of charge neutrality and beta equilibrium. In realistic nuclear matter the proton fraction is much smaller, typically around 10%, depending on the density. Its precise value is of astrophysical relevance: for example the neutrino emissivity of nuclear matter strongly depends on it since the so-called direct Urca process only becomes significant above a certain threshold for $x_P$ [1]. Here, in our prototypical approach to holographic isospin-asymmetric matter we are mostly interested in the qualitative behavior and quantitative comparisons to real-world nuclear matter are difficult. Nevertheless, it is interesting to see that even within our approach (and within the low-density approximation of this subsection) the limit of large $\lambda/N_c$ does yield arbitrarily small proton fractions, i.e., for $\lambda/N_c \to \infty$ we approach pure neutron matter, although we should keep in mind that for large $\lambda/N_c$ we are extrapolating beyond the regime of validity of holographic models.

# 3 Results: confined geometry

In this section we evaluate the model in the confined geometry and determine the preferred phases for given baryon and isospin chemical potentials (in the confined geometry, there is no temperature dependence of the phases we consider). In the practical calculation, baryon and isospin chemical potentials are treated in different ways. The simplest approach is to first fix $n_B$ and $\mu_I$. This defines the boundary conditions for $h(u)$ and $a_0(u)$ and the coupled system of equations (30b), (30c) can be solved (we have found that it is somewhat easier to transform these equations to the $z$ coordinate before solving them numerically). The resulting functions can then be used to compute the isospin number density $n_I$ from Eq. (45), the baryon chemical potential $\mu_B$ from Eq. (48), and the free energy $\Omega$ from Eq. (49). Working at a fixed $\mu_B$ is somewhat trickier because then the EOMs (30b), (30c) have to be solved simultaneously with the condition (48). In either case, the numerical evaluation can be done with standard routines in Mathematica without major difficulties.

## 3.1 Baryon and isospin densities

Let us first discuss the baryon and isospin densities as functions of $\mu_B$ for various fixed values of $\mu_I$. The results are shown in Fig. 2. In the upper panels we consider the pure baryonic phase, while pion condensation is included in the lower panels. We shall later see that the phases without pion condensate are never preferred unless $\mu_I = 0$. Nevertheless, we present the results for the pure baryonic phase as well, which is of theoretical interest but also because in a more realistic situation with nonzero pion mass we expect this phase to be important for small isospin chemical potentials.

In the upper left panel we see that for small $\mu_I$ the curves diverge to infinite $\mu_B$ for $n_B \to 0$. This was already noticed in Ref. [9] for $\mu_I = 0$, and we have observed this unphysical behavior in the low-density approximation of Sec. 2.7. It means that the current approximation does not yield a vacuum mass for the baryon, which is in contrast to the pointlike approximation for baryons [46] and the instanton gas approximation [9, 11]. As the baryon density is increased

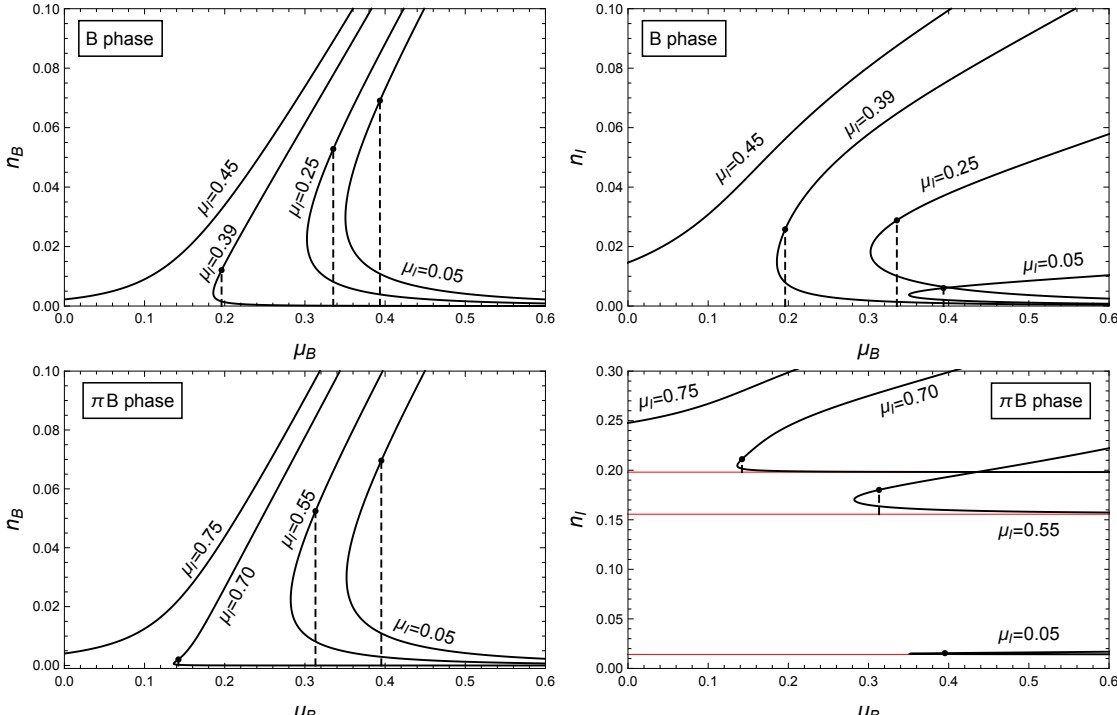

Figure 2: **Top row:** Dimensionless baryon and isospin number densities $n_B$ and $n_I$ in the pure baryonic phase as functions of $\mu_B$ at fixed values of the isospin chemical potential $\mu_I$. Dashed lines indicate first-order phase transitions from the vacuum to the B phase, i.e., the branches below the dots are metastable or unstable. **Bottom row:** Same quantities in the phase where baryonic matter coexists with a pion condensate. The thin (red) lines indicate the values of $n_I$ in the pure pion-condensed phase, and the dashed lines indicate the transition from the $\pi$ phase to the $\pi$B phase. All curves are calculated with $\lambda = 15$. The dimensionless quantities can be translated into physical units with the help of table 1 and a choice for the Kaluza-Klein scale $M_{\text{KK}}$.

the $n_B$ curves turn around and acquire a positive slope, which corresponds to the thermodynamically stable branch. By comparing free energies one finds the phase transition between the vacuum and the B phase (upper panels) and between the $\pi$ phase and the $\pi$B phase (lower panels), indicated by vertical dashed lines. We will discuss the result for the free energy itself below, see Fig. 4. The effect of the isospin chemical potential is to move the phase transition towards lower baryon chemical potentials and baryon densities, and to weaken it in the sense that the jump in the densities becomes smaller.

The most striking feature of the $n_B$ curves is that their low-density part flips from $\mu_B = +\infty$ to $\mu_B = -\infty$ at a certain critical value of $\mu_I$. This value depends on whether pion condensation is taken into account or not: we find $\mu_I \simeq 0.42$ for the critical value in the pure baryon configuration, and $\mu_I \simeq 0.71$ in the $\pi$B configuration. For $\mu_I$ larger than these critical values we see in particular that there is a nonzero baryon density even for $\mu_B = 0$. If we ignore pions, the only way for the system to create an isospin density is by creating baryons. This is exactly what the system does at sufficiently large $\mu_I$. In the presence of a pion condensate, there is already a nonzero isospin density and thus baryons are not the only way for the system to respond to the isospin chemical potential. Nevertheless, baryonic matter is created even in this situation, but now for larger values of $\mu_I$. It might seem curious that we find a positive net baryon number at $\mu_B = 0$, where there should not be any preference for baryons or antibaryons. The reason is that there also exists a "mirror" state with negative net baryon

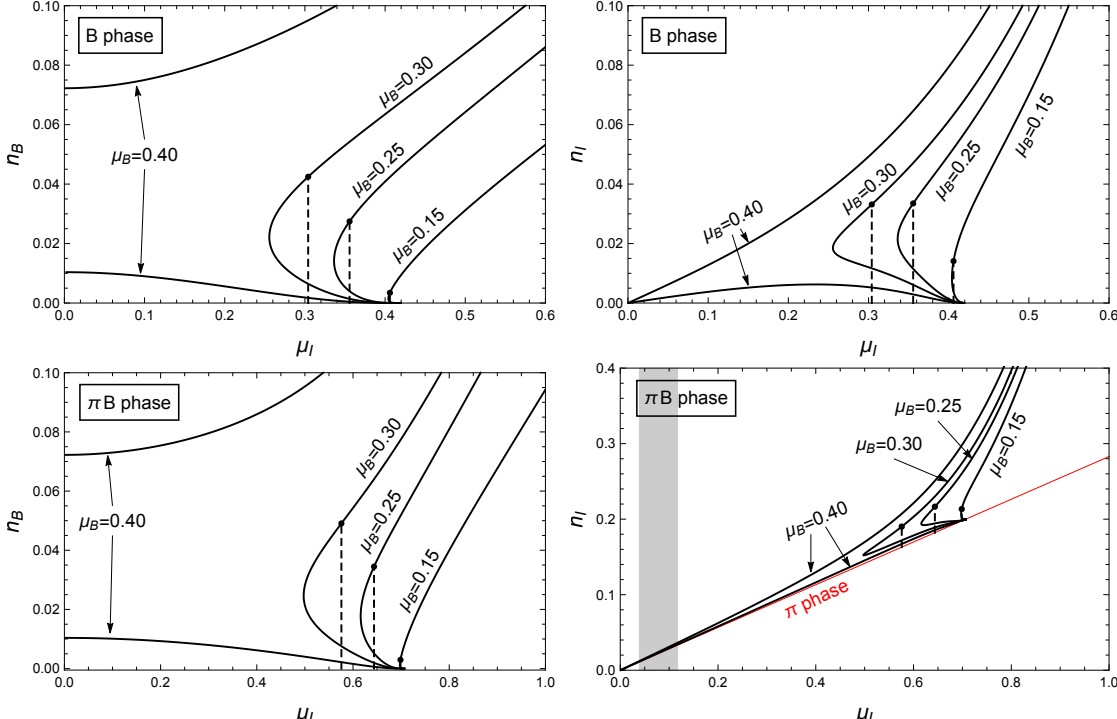

Figure 3: Counterpart to Fig. 2, now with fixed values of $\mu_B$ instead of $\mu_I$. **Top row:** Dimensionless baryon and isospin number densities $n_B$ and $n_I$ in the pure baryon phase as functions of $\mu_I$ at fixed values for the baryon chemical potential $\mu_B$. For $\mu_B = 0.40$ there are two disconnected branches, the lower one being unstable. **Bottom row:** Same quantities in the coexistence phase. The thin (red) line shows the behavior of $n_I$ in the pure pion-condensed phase, which is linear in $\mu_I$ and identical to the result from chiral perturbation theory, see Eq. (52). The gray band in the lower right panel indicates the physical pion mass $m_\pi \simeq 140\,\mathrm{MeV}$ for the range $M_{\mathrm{KK}} = (500 - 1500)\,\mathrm{MeV}$, i.e., even if the pion mass was taken into account in the calculation, pion condensation is expected to occur everywhere to the right of that band.

number with the same free energy and same isospin density, such that the symmetry between baryons and antibaryons at $\mu_B = 0$ is indeed respected. In other words, there is a first-order phase transition at large $\mu_I$ and $\mu_B = 0$ where the baryon density $n_B$ jumps from a nonzero positive value to the negative value with equal magnitude, while $n_I$ remains continuous across the transition. We shall come back to this phase transition when we discuss the phase diagram in Sec. 3.2. *A priori*, the baryon/antibaryon symmetry could also have been realized through a vanishing baryon density at $\mu_B = 0$ for all $\mu_I$. It is a prediction of our model that this is not the case and for sufficiently large $\mu_I$ a nonzero positive (negative) baryon density exists even for $\mu_B \to 0^+$ ($\mu_B \to 0^-$).

The corresponding $n_I$ curves are presented in the right panels of Fig. 2. In the B phase, the qualitative behavior of the isospin density is similar to that of the baryon density. This is consistent with the fact that in this phase the isospin density is created solely from baryons. In the $\pi$B phase, however, the results demonstrate that for vanishing $n_B$ the curves approach the nonzero value for $n_I$ of the $\pi$ phase, shown as horizontal lines. The first-order phase transition manifests itself in a jump of the isospin density from its already nonzero value to a larger value due to the onset of baryons.

In Fig. 3 we again plot the number densities $n_B$ and $n_I$, but now as functions of $\mu_I$ at various fixed values of $\mu_B$. While the general behavior at large densities is very similar to Fig. 2, now as the densities approach zero (or the density of the $\pi$ phase in the case of $n_I$ in the $\pi$B phase),

the chemical potentials remain finite. They approach asymptotic values which are exactly the values of $\mu_I$ at which the divergences in Fig. 2 flip sign, i.e., $\mu_I \simeq 0.42$ (upper panels) and $\mu_I \simeq 0.71$ (lower panels). In Fig. 3 the physical meaning of these values is more obvious. The isospin chemical potential is the energy needed to place an $N_I = 1$ charge into the system. Therefore, these critical values of $\mu_I$ can be interpreted as the mass of an $N_I = 1$ baryon placed into the vacuum (upper panels) or into a pion condensate (lower panels). Of course we need to keep in mind that these values are obtained by extrapolating our approximation, which cannot be expected to work well at low densities, to zero densities. In other words, here we are trying to make a statement about a single baryon with the help of an approximation whose starting point is a dense many-baryon system. Therefore, this interpretation has to be taken with some care. Nevertheless, it is tempting to compare our effective mass with the single-baryon result (16). Setting $N_I = 1$ in this result and using the same 't Hooft coupling $\lambda = 15$ as in the figure gives $e \simeq 0.61$, which is somewhat larger than the $N_I = 1$ vacuum mass $\mu_I \simeq 0.42$ from Fig. 3. More importantly, we observe that the effective mass of the $N_I = 1$ baryon is larger in the presence of a pion condensate compared to the vacuum. This tendency is in accordance with the arguments of Ref. [21]. There, however, it was conjectured, based on results from chiral perturbation theory (including baryons), that in QCD for $\mu_B = 0$ baryonic matter never appears as $\mu_I$ is increased. This is obviously different in our holographic model, which does go beyond chiral perturbation theory in the sense that our approximation is not expected to fail at large energies, although at asymptotically large energies our model is certainly different from QCD due to the lack of asymptotic freedom. In all curves shown in Fig. 3 we see that baryons do appear at sufficiently large $\mu_I$ through a first-order phase transition. For $\mu_B \to 0$ we see that the first-order transition becomes weaker and we have checked that at $\mu_B = 0$ the transition turns into a second-order baryon onset in both cases, i.e., no matter if we include pion condensation or not. The lower right panel shows that in the presence of pion condensation the isospin density follows the result from chiral perturbation theory until baryonic matter contributes to the isospin density. We shall come back to this behavior when we discuss non-antipodal brane configurations in the deconfined geometry, where corrections to chiral perturbation theory can be found already in the $\pi$ phase, see Fig. 8.

One might ask to what extent our conclusions will change if quark masses are taken into account. To get some idea of the effect we have added a band in the lower right panel to indicate at which point pion condensation is expected for a physical pion mass. Collecting the constants from table 1 and taking into account that in our conventions pion condensation should occur for isospin chemical potentials larger than $m_\pi/2$, the critical dimensionless isospin chemical potential is $\mu_I = m_\pi/(2\lambda_0 M_{KK})$. The limits of the band are chosen to correspond to $M_{KK} = 500\,\text{MeV}$ and $M_{KK} = 1500\,\text{MeV}$, which is a range that (generously) covers the values typically chosen for $M_{KK}$, for instance $M_{KK} = 949\,\text{MeV}$ in the original works [31, 32]. We thus conclude that all the interesting details of Fig. 3 that we have just discussed may receive corrections through quark mass effects, but are in the regime where pion condensation is expected even in the presence of a nonzero pion mass.

## 3.2 Phase structure

We have already indicated the first-order transitions to baryonic matter in the results of the previous subsection. These transitions are obtained by computing the free energy (49), which is plotted in Fig. 4 for the various candidate phases discussed in Sec. 2.6. The free energy is shown as a function of $\mu_B$ at fixed $\mu_I = 0.10$ (left panel) and as a function of $\mu_I$ at fixed $\mu_B = 0.15$ (right panel). These plots confirm that the pion-condensed phase is preferred for small nonzero isospin chemical potentials and not too large baryon chemical potentials. In this phase, the free energy is quadratic in $\mu_I$ and independent of $\mu_B$, see Eq. (52). In accordance with Figs. 2 and 3 we see that as the B and $\pi$B curves approach the non-baryonic phases, they



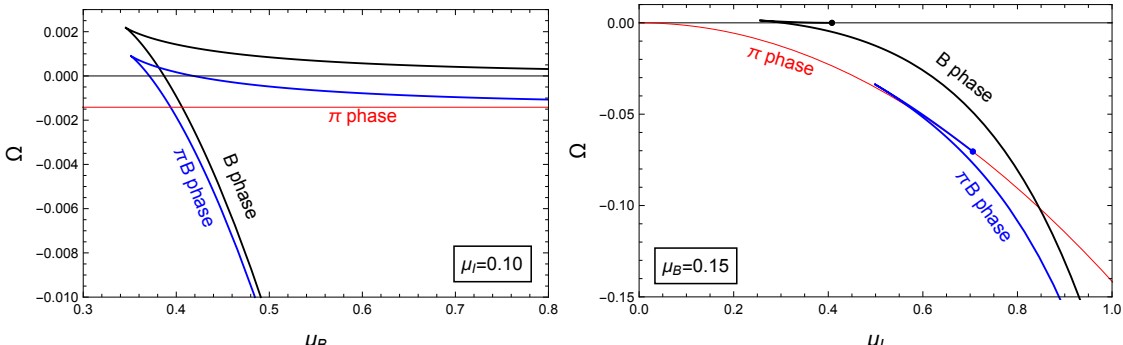

Figure 4: Dimensionless free energy density for the different phases as a function of $\mu_B$ at fixed $\mu_I = 0.10$ (left panel) and as a function of $\mu_I$ at fixed $\mu_B = 0.15$ (right panel), for $\lambda = 15$. The unlabeled thin (black) horizontal line is the vacuum $\Omega = 0$. In both cases, the ground state evolves from the pion-condensed phase via a first-order phase transition to the $\pi$B phase, where pions and baryons coexist.

go to infinite $\mu_B$ at fixed $\mu_I$, but to a finite $\mu_I$ at fixed $\mu_B$. At large $\mu_I$ and/or $\mu_B$ the phase where baryonic matter coexists with a pion condensate becomes preferred. The pure baryonic phase is never preferred. For large chemical potentials, where baryons dominate over pions, the free energies of the B phase and the $\pi$B phase approach each other.

We can now construct the phase diagram by tracing the intercept of the free energies of the $\pi$ and $\pi$B phases shown in Fig. 4 in the $\mu_I$-$\mu_B$ plane. The result is shown in the left panel of Fig. 5. Here we have also included the phase transition line in the absence of pion condensation, i.e., for the transition between the vacuum and the B phase. Strictly speaking, within our calculation in the chiral limit, this line is not part of the phase diagram since it indicates the transition between two metastable phases. Nevertheless, it is useful for comparison and also may play a role once a nonzero pion mass is included in an improved version of our setup. For $\mu_I = 0$ we have a first-order baryon onset at about $\mu_B \simeq 0.4$, as already discussed in Ref. [9]. A nonzero isospin chemical potential moves the critical $\mu_B$ to lower values, and the first-order transition becomes weaker. By comparing to the dashed curve we see that the baryon onset is delayed to larger chemical potentials by pion condensation. The phase transition line intersects the horizontal axis at $\mu_I \simeq 0.71$ (with pions) and $\mu_I \simeq 0.42$ (without pions). These are exactly the values interpreted in the previous section as effective $N_I = 1$ baryon masses. The reason why the phase transition coincides with these masses is that it has become second order in the $\mu_B \to 0$ limit.

By extending the calculation to negative baryon densities one finds that the horizontal axis is actually a first-order phase transition line beyond that second-order point, as already anticipated in the previous subsection. In other words, as one crosses the horizontal axis in the $\pi$B phase, the baryon density is discontinuous. This is manifest in the right panel of Fig. 5, where we show the phase diagram in the $n_B$-$n_I$ plane. This diagram is best understood as follows. Without pion condensation, the area enclosed by the dashed curve in the left panel shrinks to a point in the right panel because this is the vacuum where $n_B = n_I = 0$. The dashed line itself, across which the density jumps, becomes the area enclosed by the semicircle-like curve in the right panel. For densities in this area there is no stable homogeneous phase and one might construct a mixed phase of the vacuum and baryonic matter, not unlike ordinary nuclei. Finally, the first-order line along the horizontal axis also opens up to a regime where there are no "allowed" densities. In the presence of pion condensation (solid lines) the situation is similar. However, now the area enclosed by the phase transition line in the left panel shrinks to a line on the horizontal axis of the right panel (nonzero $n_I$ since this is the

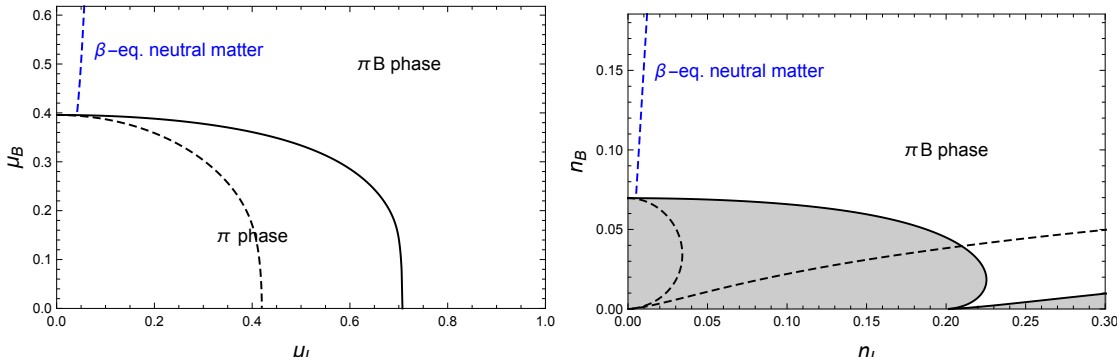

Figure 5: Phase diagram in the $\mu_I$-$\mu_B$ plane (left) and in the $n_I$-$n_B$ plane (right) for $\lambda = 15$. In both diagrams (black) solid lines are the actual phase transition lines, while the dashed lines correspond to the situation where pions are ignored, i.e., they show the phase transition from the vacuum to the B phase. The (blue) almost vertical dashed curves indicate beta-equilibrated, charge neutral matter, including leptons in the B phase. No stable homogeneous phase exists in the shaded areas in the right panel.

$\pi$ phase, not the vacuum), and the phase transition line in the left panel becomes one of the shaded areas in the right panel, where one expects a $\pi$-$\pi$B mixed phase. The second shaded area again comes from the first-order transition along the horizontal axis of the left panel.

In QCD, these phase diagrams would include chirally restored (and deconfined) matter at large $\mu_B$ and/or $\mu_I$. In the present calculation there is no further phase transition because in the confined geometry the flavor branes are necessarily connected and chiral symmetry restoration does not occur. This is one of the main reasons for us to also study the deconfined geometry, where both chirally broken and chirally restored geometries are possible, see Sec. 4.

In both panels of Fig. 5 we have indicated beta-equilibrated, charge neutral, purely baryonic matter according to Eqs. (55) and (56) (the line is dashed to emphasize that this curve is for B matter, not for $\pi$B matter). As for realistic nuclear matter in compact stars, we see that the isospin chemical potential is much smaller than the baryon chemical potential, for the curve shown here $\mu_I$ varies from about 6% to 8% of $\mu_B$. The proton fraction, however, is much larger than expected for ordinary nuclear matter under neutron star conditions. With the definition (68) we find that along the blue curve in the phase diagram $x_P \simeq 0.465$, i.e., our beta-equilibrated, charge neutral holographic baryonic matter is almost isospin symmetric. If the blue curve was continued to lower $\mu_I$ we would enter an unstable branch for which the analytical approximation (69) is valid. We have checked that our numerical result indeed approaches this value, which in this case is $x_P \simeq 0.444$, i.e., it is a good approximation also for the stable branch shown in the figure. Our large proton fraction shows that creating isospin-asymmetric baryonic matter is associated with a large energy cost in our approximation. This suggests a large symmetry energy, which we discuss next.

## 3.3 Symmetry energy

In Fig. 6 we present the symmetry energy defined in Eq. (18) for fixed $\lambda$ as a function of the baryon density (left panel) and at saturation density $n_0$ as a function of $\lambda$ (right panel). For these curves, we have calculated the derivative in Eq. (18) purely numerically. For both plots, $n_0$ is defined as the density just above the first-order baryon onset of isospin-symmetric baryonic matter. Therefore it only depends on $\lambda$, there is no difference between B and $\pi$B phases because for zero isospin asymmetry these phases are identical. For $\lambda = 15$ we have $n_0 \simeq 0.07$, see right panel of Fig. 5. The low-density symmetry energy in the B phase has a qualitative

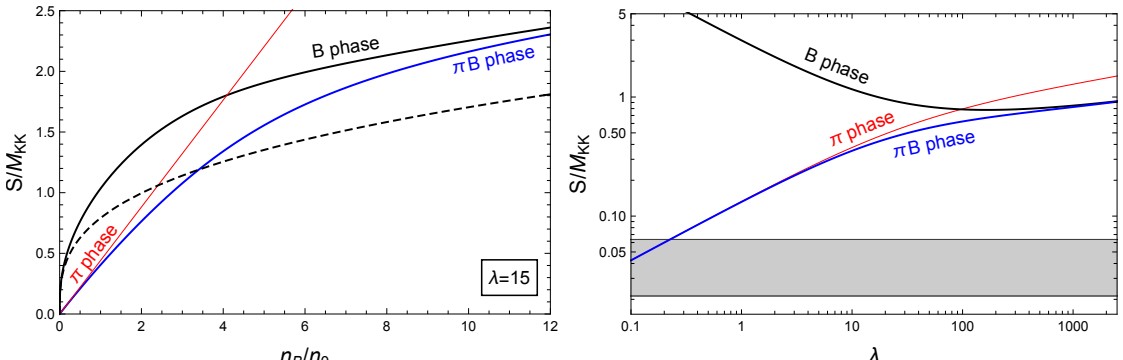

Figure 6: **Left panel:** Symmetry energy $S$ in units of $M_{KK}$ as a function of the baryon number density $n_B$ in units of the saturation density $n_0$ of isospin-symmetric baryonic matter for $\lambda = 15$ in the different phases. The unlabeled (black) dashed curve is the low-density approximation (65). **Right panel:** Symmetry energy at the ($\lambda$-dependent) saturation density $n_0$ as a function of the 't Hooft coupling $\lambda$. The shaded band indicates the physical value $S \simeq 32\,\text{MeV}$ for the range of the Kaluza-Klein scale $M_{KK} = (500 - 1500)\,\text{MeV}$.

behavior similar to many different approaches based on phenomenological models or effective theories (see for example [69] and references therein). Only for very small baryon densities our result is well approximated by the analytical approximation (65). For large densities, where the traditional approaches differ from each other [69], our symmetry energy keeps increasing monotonically, comparable to the result of a similar holographic model using a D4-D6 setup, albeit with very different approximations [63]. For comparison we have included the coexistence phase which behaves like the $\pi$ phase for small baryon densities and approaches the pure baryonic phase for large densities.

For the right panel we have calculated the saturation density for each lambda in order to take the derivative in Eq. (18) at this $\lambda$-dependent $n_0$. We observe that the symmetry energy of the B and $\pi$B phase behave very differently at small $\lambda$. As in the left panel, we see that the $\pi$B phase interpolates between the $\pi$ phase and the B phase also as a function of $\lambda$. Most importantly, this panel shows that the symmetry energy of the purely baryonic phase is much larger than the real-world value $S \simeq 32\,\text{MeV}$ [70, 71]. Namely, for any reasonable value of $M_{KK}$, for example to reproduce vacuum properties of mesons, the gray band in the right panel shows that the symmetry energy of our holographic baryonic matter is larger by an order of magnitude or more, depending on the value of $\lambda$.

This observation is in agreement with the large proton fraction observed in the previous section. The explanation of this behavior was already briefly mentioned below Eq. (16): the cold and dense isospin-asymmetric baryonic matter in our model is not made of neutrons and protons. As the single-baryon spectrum (16) suggests, we can think of our baryonic matter as a homogeneous distribution of classical instanton solutions deformed away from the usual BPST-type configuration of [36] by the presence of the isospin chemical potential. Such solutions are heavier (and effectively larger) than the isospin-symmetric ones. (The relative mass difference is a $1/\lambda$ correction such that its relative importance decreases for larger values of $\lambda$.) The crucial point is that our *isospin-symmetric* matter is different from a system with equal number of neutrons and protons, because the lightest available single-baryon state has zero isospin. Then, forcing the system to create an isospin asymmetry amounts to exciting new – heavier – baryonic states with nonzero isospin rather than simply reshuffling the occupation numbers for neutron and proton states, resulting in a much larger symmetry energy. The difference between a system of protons and neutrons and a gas of such deformed classical solutions was also discussed in the context of the Skyrme model in Ref. [72], where it was argued

that the classical solutions are more accurate approximations for larger rather than smaller isospin asymmetries (the symmetry energy, where the discrepancy of our results to real-world nuclear matter is most obvious, is a derivative evaluated at vanishing isospin asymmetry). Analogous considerations hold in our context since, as shown in Ref. [36], the states with different isospin eigenvalues are obtained in the Witten-Sakai-Sugimoto model by quantizing the moduli space of slowly moving instantons in analogy with the corresponding procedure for Skyrmions [73]. It would be very interesting to construct isospin-asymmetric dense matter configurations starting from the holographic protons and neutrons of Ref. [36], perhaps along the lines of the instanton gas in [9].

## 4 Deconfined geometry

The setting of the confined geometry and maximally separated flavor branes of the previous sections was well suited to explain our main ideas and for a systematic evaluation without significant numerical difficulties. For a better applicability to real-world QCD it is desirable to perform the analogous calculation in the deconfined geometry. This allows us to include temperature effects and the possibility of chiral restoration. The price we have to pay is a more involved calculation which also poses some numerical difficulties in the evaluation. We shall therefore, after deriving the relevant EOMs and thermodynamic quantities, be less exhaustive in the evaluation and restrict ourselves to a few key results.

### 4.1 Setup

In the deconfined geometry, the induced metric on (half of) the D8-$\overline{\text{D8}}$ flavor branes is

$$ds^2 = \left(\frac{U}{R}\right)^{3/2} (f_T dX_0^2 + d\mathbf{X}^2) + \left(\frac{R}{U}\right)^{3/2} \left\{ \left[\frac{1}{f_T} + \left(\frac{U}{R}\right)^3 (\partial_U X_4)^2\right] dU^2 + U^2 d\Omega_4^2 \right\}, \quad (70)$$

where

$$f_T(U) = 1 - \frac{U_T^3}{U^3}, \qquad U_T = \left(\frac{4\pi}{3}T\right)^2 R^3, \quad (71)$$

such that in our conventions the dimensionless temperature is $t = 3u_T^{1/2}/(4\pi)$. Moreover, the function $X_4(U)$ describes the shape of the flavor branes. This setup corresponds to the high-temperature phase of the background geometry, usually associated with the deconfined phase of the dual field theory (see however Ref. [74]). Its topology is such that the flavor branes are allowed to extend all the way to the black hole horizon. Thus, whether they join in the bulk or not becomes a dynamical question and depends in particular on their asymptotic separation $X_4(U \to \infty) = L/2$. We may think of $L$ (or its dimensionless counterpart $\ell = LM_{\text{KK}}$) as a third model parameter besides $\lambda$ and $M_{\text{KK}}$. This extension produces new interesting physics compared to the antipodal case $\ell = \pi$. In particular, it allows for the appearance of a deconfined but chirally broken phase [75], such that the chiral transition depends on the chemical potentials $\mu_B$ and $\mu_I$, as expected in QCD at $N_c = 3$. Following Refs. [9,11,12,38, 39,60], we choose to work in the so-called "decompactified" limit, characterized by a small separation $\ell \ll \pi$. In this limit, glue and flavor physics become decoupled, and we employ the metric (70) for arbitrarily small temperatures. Since the gluon dynamics is neglected in this approach, the dual field theory bears resemblance to the Nambu-Jona-Lasinio model [39,76,77]. Some of the top-down control is lost in this limit since, strictly speaking, the Kaluza-Klein modes become relevant. Nevertheless, this effective approach has proven to yield very interesting insights akin to a much richer phase structure, see for example the recent study

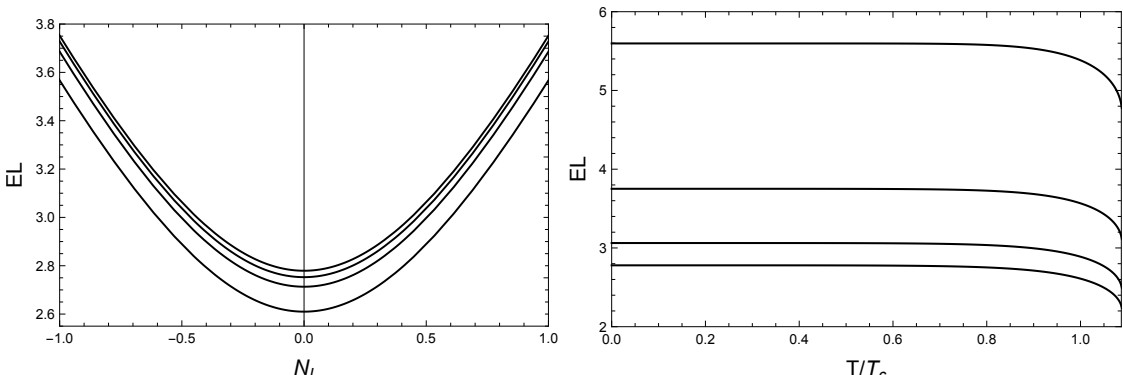

Figure 7: Single-baryon energy as a function of the isospin number for different temperatures (left panel), $T/T_c = 0, 0.8, 0.9, 1$ from top to bottom, and as a function of temperature for different isospin numbers (right panel), $N_I = 0, 0.5, 1, 2$ from bottom to top. The energy is given in units of the inverse asymptotic separation of the flavor branes $L^{-1}$, while $T_c$ is the critical temperature for the chiral phase transition, and we have set $\lambda/\ell = 20$. In the semi-classical approximation of this paper, the spectrum is continuous in $N_I$.

of holographic quarkyonic matter [12]. We emphasize that, besides the fact that we use Eq. (70) for all $t$, the small $\ell$ limit is not enforced explicitly in any of the following calculations.

The action $S$ is again given by a DBI and a CS term as in Eq. (3). Since the CS term does not depend on the metric, it has the same form as in the confined geometry. The DBI action is, in analogy to Eq. (4),

$$S_{\text{DBI}} = 2T_8 V_4 \int d^4X \int_{U_c}^{\infty} dU e^{-\phi} \, \text{STr} \sqrt{\det(g + 2\pi\alpha'\mathcal{F})}, \tag{72}$$

where $U_c > U_T$ is the value of the holographic coordinate at the tip of the connected flavor branes in the chirally broken phase. This value will have to be determined dynamically and depends on temperature and the chemical potentials. In the chirally restored phase, the branes are straight, $X_4' = 0$, and $U \in [U_T, \infty]$ instead. As already briefly discussed below Eq. (4), it is necessary to state precisely how the action (72) is to be interpreted in the non-abelian case. Although the exact answer is not known, a useful prescription was put forward in Ref. [78] and used in a context similar to ours for instance in Refs. [10,79]. The idea is to first compute the determinant as if the gauge fields were abelian, which yields (using the same dimensionless quantities as in Sec. 2)

$$S_{\text{DBI}} = \frac{\mathcal{N}}{\lambda_0 M_{\text{KK}}^4} \int d^4x \int_{u_c}^{\infty} du \, \mathcal{L}_{\text{DBI}}, \tag{73}$$

with $\mathcal{N}$ defined in Eq. (27), and where the DBI Lagrangian is

$$\mathcal{L}_{\text{DBI}} = u^{5/2} \, \text{STr} \left\{ f_T \frac{\mathcal{F}_{iu}^2}{\lambda_0^2} + (1 + u^3 f_T x_4'^2 + \mathcal{F}_{0u}^2) \left( 1 + \frac{\mathcal{F}_{ij}^2}{2u^3 \lambda_0^2} \right) + \frac{f_T (\mathcal{F}_{ij} \mathcal{F}_{ku} \epsilon_{ijk})^2}{4u^3 \lambda_0^4} \right.$$

$$\left. + \frac{1 + u^3 f_T x_4'^2}{u^3 f_T} \left[ \mathcal{F}_{0i}^2 + \frac{(\mathcal{F}_{ij} \mathcal{F}_{k0} \epsilon_{ijk})^2}{4u^3 \lambda_0^2} \right] + \frac{\mathcal{F}_{0i}^2 \mathcal{F}_{ju}^2 - (\mathcal{F}_{0i} \mathcal{F}_{iu})^2 + 2\mathcal{F}_{0u} \mathcal{F}_{0i} \mathcal{F}_{ij} \mathcal{F}_{ju}}{u^3 \lambda_0^2} \right\}^{1/2}. \tag{74}$$

Now, firstly, we again consider the energy of a single baryon with isospin, in analogy to the confined geometry, see Sec. 2.2. To this end, we work with the simple YM Lagrangian and the BPST instanton solution, which is a good approximation for large $\lambda$. This calculation is carried

out in appendix A and leads to the (dimensionless) energy

$$e = u_c \frac{\sqrt{f_T(u_c)}}{3} + \frac{\sqrt{6\beta_0 u_c}}{\lambda_0} \sqrt{\frac{N_I^2}{6} + \frac{2}{15}}, \tag{75}$$

where we have abbreviated

$$\beta_0 \equiv 1 - \frac{u_T^3}{8u_c^3} - \frac{5u_T^6}{16u_c^6}. \tag{76}$$

For vanishing isospin we recover the result of Ref. [10]. The energy is very similar to the one of the confined case (16). In particular, we observe the same dependence on the isospin number $N_I$. The main difference is the temperature dependence. The temperature enters not only in $u_T$ but also through $u_c$, which has to be calculated numerically for each temperature. Since we have derived Eq. (75) from putting a single baryon into the mesonic vacuum (ignoring pion condensation), $u_c$ has to be determined in that phase, for instance using the equations given for the vacuum in Sec. 4.3 below. In Fig. 7 we plot the baryon mass $E = \lambda_0 N_c M_{KK} e$ as a function of $N_I$ for fixed temperatures (left) and as a function of $T$ for fixed isospin numbers (right). In this figure, $T_c$ is the critical temperature for the chiral phase transition, obtained by comparing the free energies of the vacuum with the chirally restored phase, which is also discussed in Sec. 4.3. The behavior of the baryon mass, being almost constant in $T$ before it decreases as we approach $T_c$ is similar to thermal baryon masses calculated on the lattice [80].

Secondly, our main focus is again on the thermodynamic system with nonzero baryon and isospin densities. As in the confined case, we shall use the homogeneous ansatz (20) and the resulting field strengths (21). In principle, one can expand the square root in Eq. (74) and take the symmetrized trace of each individual term. This prescription is known to be consistent with open string theory amplitudes up to $\mathcal{O}(F^6)$ corrections. Within our ansatz this procedure can be carried out, and the expansion can be resummed explicitly, as we demonstrate in appendix B. However, the all-order result is much too complicated to be of practical use for our purposes. Truncations of the resulting infinite series at $\mathcal{O}(F^2)$ or $\mathcal{O}(F^4)$ are possible, but also lead to a relatively complicated action due to the presence of the embedding function. We circumvent these complications by using the following action (including the CS contribution),

$$S = \mathcal{N} N_f \frac{V}{T} \int_{u_c}^{\infty} du \, \mathcal{L}, \tag{77}$$

where

$$\mathcal{L} = u^{5/2} \sqrt{(1 + u^3 f_T x_4'^2 + g_1 - \hat{a}_0'^2 - a_0'^2)(1 + g_2 - g_3)} - \frac{9}{4} \lambda_0 \hat{a}_0 h^2 h', \tag{78}$$

with

$$g_1 \equiv \frac{3 f_T h'^2}{4}, \qquad g_2 \equiv \frac{3 \lambda_0^2 h^4}{4u^3}, \qquad g_3 \equiv \frac{2 \lambda_0^2 h^2 a_0^2}{u^3 f_T}. \tag{79}$$

These functions differ from their counterparts in the confined geometry (29) due to the different metric. (In a slight abuse of notation we use the same symbols for them, but since the confined and deconfined calculations are clearly separated this should not lead to any confusion.)

The reasons for our approximation (78) are as follows. To $\mathcal{O}(F^2)$ we reproduce the YM approximation, which in turn is identical to the truncated result from the symmetrized trace prescription carried out in appendix B. (Our approximation (78) does *not* yield the $\mathcal{O}(F^4)$ result from that prescription.) The isospin asymmetric terms are included in a simple way, motivated by how they enter the YM Lagrangian in the confined case (28). In the isospin-symmetric limit $a_0 = g_3 = 0$, we recover the Lagrangian of Ref. [9], while retaining the simplifications due to

the factorized square root structure, which, as we shall see, allows for a trivial first integration of the EOMs. We will also be able to compute relatively simple semi-analytical expressions for the functions $x_4(u)$ and $\hat{a}_0(u)$, and in the general method we subsequently use for solving the system we can then follow Ref. [9]. Had we used the YM approximation, which can be obtained by expanding the square root in Eq. (78) to second order in the field strengths, the resulting expressions would have been much more complicated.

## 4.2 Equations of motion and free energy

The procedure for solving the EOMs and computing the free energy density is conceptually analogous to but technically more involved than that of Sec. 2, mainly because of the embedding function $x_4(u)$ and the associated dynamical parameter $u_c$. In this subsection, we focus on the chirally broken configurations, i.e., we assume the D8-$\overline{\text{D8}}$ pairs to join in the bulk. For this scenario we define the coordinate $z \in [-\infty, \infty]$ in analogy to Eq. (22),

$$u^3 = u_c^3 + u_c z^2, \tag{80}$$

and we will again make use of both coordinates, depending on which one is more convenient for a given calculation or argument.

The integrated EOMs for $x_4$ and $\hat{a}_0$ are

$$\frac{u^{5/2}\hat{a}_0'\sqrt{1 + g_2 - g_3}}{\sqrt{1 + u^3 f_T x_4'^2 + g_1 - \hat{a}_0'^2 - a_0'^2}} = n_B Q, \tag{81a}$$

$$\frac{u^{11/2} f_T x_4' \sqrt{1 + g_2 - g_3}}{\sqrt{1 + u^3 f_T x_4'^2 + g_1 - \hat{a}_0'^2 - a_0'^2}} = k, \tag{81b}$$

where $Q = 1 - h^3/h_c^3$ as defined in Eq. (31), and $k$ is an integration constant to be determined below. We can solve these equations algebraically for $x_4'$ and $\hat{a}_0'$ and write the result compactly as

$$\hat{a}_0' = \frac{n_B Q}{u^{5/2}}\zeta, \qquad x_4' = \frac{k}{u^{11/2} f_T}\zeta, \tag{82}$$

where we have abbreviated

$$\zeta \equiv \frac{\sqrt{1 + g_1 + u^3 f_T x_4'^2 - \hat{a}_0'^2 - a_0'^2}}{\sqrt{1 + g_2 - g_3}} = \frac{\sqrt{1 + g_1 - a_0'^2}}{\sqrt{1 + g_2 - g_3 + \frac{(n_B Q)^2}{u^5} - \frac{k^2}{u^8 f_T}}}. \tag{83}$$

Using this abbreviation and the solutions (82), the EOMs for $a_0$ and $h$ read

$$\partial_u \left( \frac{u^{5/2} a_0'}{\zeta} \right) = \frac{2\lambda_0^2 h^2 a_0}{u^{1/2} f_T}\zeta, \tag{84a}$$

$$\partial_u \left( \frac{u^{5/2} f_T h'}{\zeta} \right) - \frac{3\lambda_0^2 h^2 n_B Q}{u^{5/2}}\zeta = \frac{2\lambda_0^2 h\zeta}{3u^{1/2}} \left( 3h^2 - \frac{4a_0^2}{f_T} \right). \tag{84b}$$

As in Sec. 2, the function $h$ is discontinuous at $u = u_c$, and its IR boundary condition is given by the baryon density, see Eq. (25). The UV boundary conditions are the same as in the confined

geometry, see table 2. It is convenient to rewrite the boundary conditions for the embedding function and the temporal components of the gauge fields as

$$\frac{\ell}{2} \;=\; \int_{u_c}^{\infty} du\, x_4' , \tag{85a}$$

$$\mu_B \;=\; \int_{u_c}^{\infty} du\, \hat{a}_0' + \hat{a}_0(u_c) , \tag{85b}$$

$$\mu_I \;=\; \int_{u_c}^{\infty} du\, a_0' + a_0(u_c) . \tag{85c}$$

As in the confined geometry we find two types of solutions, depending on whether the non-abelian component $a_0(z)$ is symmetric or antisymmetric under $z \to -z$, which is determined by the type of boundary conditions. Once again, it is useful to introduce the coefficients of the expansions around the tip of the connected branes $z = 0$, which corresponds to $u = u_c$. We use the same notation as in the deconfined geometry, i.e., the functions $a_0$, and $h$ have the expansions (39) with $u_{\mathrm{KK}}$ replaced by $u_c$, and the same continuations to the second half of the connected branes as explained below these expansions. With the help of these expansions we find

$$\zeta = \frac{c}{\sqrt{u - u_c}} + \dots = \frac{\sqrt{3 u_c}\, c}{z} + \dots , \tag{86}$$

with the abbreviation

$$c \equiv \frac{1}{4} \sqrt{ \frac{3 f_T(u_c) h_{(1)}^2 - 4 a_{(1)}^2}{1 - \frac{k^2}{u_c^8 f_T(u_c)} + \frac{\lambda_0^2 h_c^2}{4 u_c^3}\left[3 h_c^2 - \frac{8 a_c^2}{f_T(u_c)}\right]} } . \tag{87}$$

With $x_4'$ from Eq. (82) this result implies that $x_4'$ diverges at $u = u_c$, and thus the brane embedding is smooth, even in the presence of the discontinuity in $h$. This result is valid for both types of boundary conditions for $a_0$. In the symmetric case ($\sigma$-type boundary conditions) the coefficient of the linear term vanishes, $a_{(1)} = 0$, while in the anti-symmetric case ($\pi$-type boundary conditions) the value at $u = u_c$ vanishes, $a_c = 0$.

The dimensionless free energy density is

$$\Omega = \int_{u_c}^{\infty} du\, \mathcal{L} = \frac{1}{2} \int_{-\infty}^{\infty} dz\, \frac{\partial u}{\partial z} \mathcal{L} , \tag{88}$$

with the Lagrangian $\mathcal{L}$ from Eq. (78) evaluated at the stationary point. In analogy to Eq. (42), we write the derivatives of the free energy with respect to $x = \mu_B, \mu_I, h_c, a_c$ (with the other of these variables held fixed) as

$$\frac{\partial \Omega}{\partial x} \;=\; \frac{1}{2} \int_{-\infty}^{\infty} dz \left\{ -\partial_z \left( \frac{u^{5/2} \hat{a}_0'}{\zeta} \frac{\partial \hat{a}_0}{\partial x} \right) - \partial_z \left( \frac{u^{5/2} a_0'}{\zeta} \frac{\partial a_0}{\partial x} \right) + \partial_z \left( k \frac{\partial x_4}{\partial x} \right) \right.$$

$$\left. + \partial_z \left[ \left( \frac{3 u^{5/2} f_T h'}{4 \zeta} - \frac{9 \lambda_0 \hat{a}_0 h^2}{4} \right) \frac{\partial h}{\partial x} \right] \right\} , \tag{89}$$

where we have used Eq. (82). For the first two terms we need $\zeta(z = \pm\infty) = 1$, and the second term creates a nonzero contribution from $z = 0$ if, for now, we allow $a_0'$ to be discontinuous. The third term vanishes since the boundary value of $x_4$ is a fixed model parameter. Finally, in

the fourth term we need to take into account the discontinuity of $h$ at $z = 0$. We thus obtain (going back to the formulation in terms of the coordinate $u$)

$$\frac{\partial \Omega}{\partial x} = -n_B \frac{\partial \mu_B}{\partial x} - (u^{5/2} a_0')_\infty \frac{\partial \mu_I}{\partial x} + \frac{u_c^{5/2} a_{(1)}}{2c} \frac{\partial a_c}{\partial x} + \left[ \frac{3 u_c^{5/2} f_T(u_c) h_{(1)}}{8c} - \frac{9 \lambda_0 \hat{a}_c h_c^2}{4} \right] \frac{\partial h_c}{\partial x} . \quad (90)$$

With $x = \mu_B$ we simply confirm the usual thermodynamic relation between baryon chemical potential and baryon density, i.e., the baryon density is indeed given by the boundary condition for $h(u)$, also in the thermodynamic sense. Then, we use $x = \mu_I$ to identify the isospin density, which, with the help of Eq. (84a) can be written as

$$n_I = (u^{5/2} a_0')_{u \to \infty} = \frac{u_c^{5/2} a_{(1)}}{2c} + 2\lambda_0^2 \int_{u_c}^\infty du \frac{h^2 a_0}{u^{1/2} f_T} \zeta . \quad (91)$$

Next, requiring the free energy to be stationary with respect to $x = h_c$ yields

$$\begin{aligned}
\hat{a}_c &= \frac{u_c^{5/2} f_T(u_c) h_{(1)}}{6 c \lambda_0 h_c^2} \\
&= \frac{u_c \sqrt{f_T(u_c)}}{3} \frac{h_{(1)}}{\sqrt{h_{(1)}^2 - \frac{4 a_{(1)}^2}{3 f_T(u_c)}}} \sqrt{1 - \frac{8 a_c^2}{3 h_c^2 f_T(u_c)} + \frac{4 u_c^3}{3 \lambda_0^2 h_c^4} \left[ 1 - \frac{k^2}{u_c^8 f_T(u_c)} \right]} . \quad (92)
\end{aligned}$$

This relation is needed to compute $\mu_B$ from the numerical solutions with the help of Eq. (85b). The explicit expression on the right-hand side is interesting because in the absence of isospin, $a_c = a_{(1)} = 0$, together with the limit of large 't Hooft coupling, $\lambda_0 \to \infty$, it reduces to the vacuum mass of the baryon in the pointlike limit. This connection between the homogeneous ansatz and the completely different pointlike approach, which is based on the instanton picture, was already pointed out in Ref. [50], see also Eq. (73) in Ref. [9]. It is not obvious how to generalize the pointlike approximation to nonzero isospin. If $\hat{a}_c$ can still be interpreted as the baryon mass, Eq. (92) – in the limit $\lambda_0 \to \infty$, but keeping $a_c$ and $a_{(1)}$ nonzero – might be helpful to develop such a generalization because it contains the isospin corrections to the mass of a pointlike baryon with ($a_c = 0$) and without ($a_{(1)} = 0$) pion condensation. Finally, the conclusion from Eq. (90) for $x = a_c$ is the same as in the confined case: for $\sigma$-type boundary conditions we obtain the smoothness condition $a_{(1)} = 0$, while for $\pi$-type conditions we need to impose the additional boundary condition $a_c = 0$.

The free energy should also be minimized by the value of $u_c$. The derivative with respect to this parameter is best done separately because one has to be more careful in the derivation, as pointed out in Ref. [9]. Since the IR boundary values depend on $u_c$ not only through $u$ but also explicitly, we write

$$\left. \frac{\partial x_4}{\partial u_c} \right|_{u = u_c} = \frac{\partial x_4(u_c)}{\partial u_c} - x_4'(u_c) , \quad (93)$$

and analogously for $\hat{a}_0$, $a_0$ and $h$. Starting from the formulation in the $u$ coordinate for the $z > 0$ half, we find

$$\begin{aligned}
\frac{\partial \Omega}{\partial u_c} &= \left[ k x_4' - n_B Q \hat{a}_0' - u^{5/2} \zeta^{-1} a_0'^2 + \left( \frac{3 u^{5/2} f_T h'}{4\zeta} - \frac{9 \lambda_0 \hat{a}_0 h^2}{4} \right) h' - \mathcal{L} \right]_{u = u_c} \\
&= -\left. \frac{u^{5/2}}{\zeta} \right|_{u = u_c} = 0 , \quad (94)
\end{aligned}$$

where we have used Eqs. (78), (79), (82), and (83). We see that the minimization with respect to $u_c$ is equivalent to the smoothness of $x_4$ and is automatically satisfied, as already noticed in the absence of an isospin asymmetry [9].

We can use partial integration and the EOMs to derive a useful form of the free energy at the stationary point. In contrast to the YM approximation that we used in Sec. 2, now the free energy is formally divergent. We subtract the medium-independent term $\frac{2}{7}\Lambda^{7/2}$, where $\Lambda$ is a UV cutoff, and the resulting renormalized free energy density can be written as

$$\Omega = \int_{u_c}^{\infty} du \left( \frac{1+g_1}{\zeta} + g_3\zeta - 1 \right) - \frac{2}{7}u_c^{7/2} - n_B\mu_B - n_I\mu_I + k\frac{\ell}{2} , \tag{95}$$

for both types of boundary conditions, where the integral is now manifestly finite.

## 4.3 Possible phases

As in the confined geometry, our setup allows us to discuss and compare different types of solutions, corresponding to distinct physical phases. Here we also need to take into account the chirally restored phase, where the flavor branes are straight. The chirally broken phases are analogous to those obtained in the confined case, see section 2.6. These phases can all be obtained as limits of our general expressions of the previous subsection. We now list all phases we consider.

- **Vacuum:** The vacuum contains neither pions nor baryons, i.e., here we set $h = 0$ and use $\sigma$-type boundary conditions. This yields the constant gauge fields $\hat{a}_0(u) = \mu_B$, $a_0(u) = \mu_I$. One also finds $k = u_c^4\sqrt{f_T(u_c)}$, and the embedding function is given by

$$x_4'^2 = \frac{u_c^8 f_T(u_c)}{u^3 f_T(u) \left[ u^8 f_T(u) - u_c^8 f_T(u_c) \right]} , \tag{96}$$

with $u_c$ computed from the boundary condition (85a). The renormalized free energy is independent of the chemical potentials and takes the form

$$\Omega = \int_{u_c}^{\infty} du\, u^{5/2} \left\{ \left[ 1 - \frac{u_c^8 f_T(u_c)}{u^8 f_T} \right]^{-1/2} - 1 \right\} - \frac{2}{7}u_c^{7/2} . \tag{97}$$

At zero temperature one obtains the analytic expressions

$$u_c = \frac{16\pi^2}{\ell^2} \left[ \frac{\Gamma(9/16)}{\Gamma(1/16)} \right]^2 , \qquad \Omega = -\frac{2^{15}\pi^4}{15\ell^7} \tan\left( \frac{\pi}{16} \right) \frac{\Gamma(31/16)}{\Gamma(23/16)} \left[ \frac{\Gamma(9/16)}{\Gamma(1/16)} \right]^7 . \tag{98}$$

- **Pion-condensed phase:** In this phase, the baryon density is zero, and thus $h = 0$. As a consequence, the properties of this phase do not depend on $\mu_B$, and $\hat{a}_0(u)$ is constant, as in the vacuum. Due to the $\pi$-type boundary conditions, however, $a_0(u)$ is nontrivial and creates an isospin density $n_I$. In contrast to the confined geometry, $a_0(u)$ does not have a simple analytical form. Integrating its EOM and the one for $x_4(u)$ gives

$$a_0' = \frac{n_I}{u^{5/2}}\zeta , \qquad x_4' = \frac{k}{u^{11/2}f_T}\zeta , \tag{99}$$

where

$$\zeta = \frac{1}{\sqrt{1 + \frac{n_I^2}{u^5} - \frac{k^2}{u^8 f_T}}} , \qquad k = u_c^4 \sqrt{f_T(u_c)\left( 1 + \frac{n_I^2}{u_c^5} \right)} . \tag{100}$$

For given $\ell$ and $\mu_I$, we can then determine $u_c$ and $n_I$ from the boundary conditions (85a) and (85c) with $a_0(u_c) = 0$. These conditions have to be solved numerically, and the results can be inserted into the renormalized free energy

$$\Omega = \int_{u_c}^{\infty} du\, u^{5/2}\left(\frac{1}{\zeta} - 1\right) - \frac{2}{7}u_c^{7/2} - n_I\mu_I + k\frac{\ell}{2}. \tag{101}$$

Moreover, for small isospin densities we can derive an analytical solution. To lowest order in $n_I$ we may set $n_I = 0$ in $\zeta$ and obtain from (85a) and (85c)

$$n_I \simeq \frac{8\mu_I}{\ell^3}\left(\int_1^{\infty}\frac{du}{u^{3/2}\sqrt{u^8 - 1}}\right)^3\left(\int_1^{\infty}\frac{du\, u^{3/2}}{\sqrt{u^8 - 1}}\right)^{-1}. \tag{102}$$

After performing the integrals and inserting the relevant constants to translate our dimensionless quantities into physical ones, this relation reads

$$\left(\frac{N_c N_f \lambda_0^2 M_{KK}^3}{6\pi^2}n_I\right) \simeq 4f_{\pi}^2(\lambda_0 M_{KK}\mu_I), \tag{103}$$

with the pion decay constant in the deconfined geometry [12, 81],

$$f_{\pi}^2 = \frac{32\lambda N_c M_{KK}^2}{3\pi^2\ell^3}\left(\frac{\Gamma[9/16]}{\Gamma[1/16]}\right)^3\frac{\Gamma[11/16]}{\Gamma[3/16]}. \tag{104}$$

The relation between isospin density and isospin chemical potential (103) is thus in exact agreement with chiral perturbation theory in the limit of vanishing pion mass. This was already observed in the confined phase, see Eq. (53). However, in that case the result was exact. Interestingly, in the deconfined setting there are corrections to this relation at larger values of $n_I$ (even without including baryons), as will become apparent in the next subsection, see Fig. 8.

- **Pure baryonic phase:** In this case we allow for the presence of baryons, and we work with $\sigma$-type boundary conditions, such that there is no pion condensate. The numerical procedure for solving the EOMs is somewhat involved. First, we write the integration constant $k$ in terms of the coefficients of the expansions around $u = u_c$ by demanding the EOMs (84) to be fulfilled order by order in $u - u_c$. The order $(u - u_c)^{-1/2}$ contribution of Eq. (84a) vanishes for

$$a_{(2)} = \frac{4c^2\lambda_0^2 h_c^2 a_c}{u_c^3 f_T(u_c)}, \tag{105}$$

with $c$ defined in Eq. (87). Taking this expression for $a_{(2)}$ into account, the order $(u - u_c)^0$ contribution of Eq. (84b) yields the desired expression for $k$,

$$k^2 = \frac{u_c^8 f_T(u_c)}{16u_c + 9h_{(1)}^2 f_T(u_c)}\left\{16u_c - 3h_{(1)}^2\left(5 - 2\frac{u_T^3}{u_c^3}\right)\right.$$

$$\left. + \frac{3\lambda_0^2 h_c^4}{4u_c^3}\left[16u_c - 3h_{(1)}^2\left(2 + \frac{u_T^3}{u_c^3}\right)\right] - \frac{4\lambda_0^2 h_c^2 a_c^2}{u_c^3 f_T(u_c)}\left[8u_c - 3h_{(1)}^2 f_T(u_c)\right]\right\}. \tag{106}$$

This result can now be inserted back into the EOMs (84), which, then, are coupled differential equations for $h(u)$ and $a_0(u)$ that contain the unknown coefficients $h_{(1)}$ and $a_c$ explicitly. In the simplest setting $n_B$ and $\mu_I$ are given, which determines the boundary

conditions $h(u_c) = h_c$ and $a_0(\infty) = \mu_I$, respectively. In addition, the equations contain the unknown parameter $u_c$. Therefore, we have to solve them simultaneously with the condition (85a). This can be done with the help of the shooting method. As in the confined case, if we work at fixed $\mu_B$ instead, the additional equation (85b) together with the expression (92) for $\hat{a}_c$ has to be added to this system of equations. In fact, this is what we do to obtain the results of the following subsection, where we discuss the system at $\mu_B = 0$. In either case, we observe that the (fixed) parameter $\ell$ drops out of all equations after an appropriate rescaling of all variables, which is given in table II of Ref. [9]. We thus do not have to choose a value for $\ell$ before the numerical evaluation and rather can reinsert the appropriate powers of $\ell$ after the calculation. It turns out to be useful to employ a similar rescaling with the (dynamical) parameter $u_c$, also given in table II of Ref. [9]. This further simplifies the numerical problem, although it does not decouple any of the equations (it would completely eliminate $u_c$ from the EOMs if also the 't Hooft coupling $\lambda$ and the temperature $t$ were rescaled appropriately, but this would not allow us to work at fixed $\lambda$ and $t$). Also, as for the confined geometry, we find that the numerical evaluation is best done in the $z$ coordinate. The calculation then yields $h(u)$, $a_0(u)$, $u_c$, from which we extract $h_{(1)}$ and $a_c$, and use all this to compute the remaining thermodynamic quantities, in particular the free energy via Eq. (95). The entire calculation can be done using Mathematica, but the numerics turn out to be much more time consuming than in the confined phase with antipodally separated flavor branes.

- **Coexistence phase:** In this phase baryonic matter coexists with a pion condensate, which is taken into account by imposing $\pi$-type boundary conditions. As a result, the isospin density $n_I$ receives an extra contribution from the boundary term in Eq. (91). Again, we first need to determine the integration constant $k$. Now, both EOMs (84a) and (84b) are fulfilled to order $(u-u_c)^{-1/2}$ if

$$a_{(2)} = 0, \qquad h_{(2)} = \frac{4c^2\lambda_0^2 h_c^3}{u_c^3 f_T(u_c)}. \tag{107}$$

Then, to order $(u-u_c)^0$ the EOMs can only be satisfied if

$$
\begin{aligned}
k^2 &= \frac{u_c^8 f_T(u_c)}{16u_c + 9h_{(1)}^2 f_T(u_c) - \frac{12a_{(1)}^2}{f_T(u_c)}} \left\{ 16u_c + 20a_{(1)}^2 - 3h_{(1)}^2 \left(5 - 2\frac{u_T^3}{u_c^3}\right) \right. \\
&\quad \left. + \frac{3\lambda_0^2 h_c^4}{4u_c^3} \left[ 16u_c - 3h_{(1)}^2 \left(2 + \frac{u_T^3}{u_c^3}\right) + 8a_{(1)}^2 \right] \right\}.
\end{aligned}
\tag{108}
$$

With the help of these relations the procedure is analogous to that of the purely baryonic phase: For given $n_B$ and $\mu_I$ we compute $h(u)$, $a_0(u)$, $u_c$, which give $h_{(1)}$ and $a_{(1)}$, and $\mu_B$, $n_I$, $\Omega$ can be computed from these results.

- **Chirally symmetric phase:** Finally, in the chirally symmetric phase the flavor branes are straight, $x_4' = 0$, and extend all the way down to the horizon at $u = u_T$. Here we set the "baryon field" $h$ to zero[5]. We thus have two independent sets of gauge fields, and

---

[5]It is conceivable that baryons exist as an ingredient of chirally symmetric *quarkyonic* matter, which was discussed within the pointlike approximation and found to be preferred at large baryon densities [12]. Whether this phase can be constructed within our current ansatz is beyond the scope of this paper. Also, we restrict ourselves to the case where the branes of both flavors are straight, although phases with one connected and one straight pair of branes are conceivable as well [82].

may simply work with one half of the configuration, imposing the boundary conditions

$$\hat{a}_0(\infty) = \mu_B, \qquad a_0(\infty) = \mu_I, \qquad \hat{a}_0(u_T) = a_0(u_T) = 0. \tag{109}$$

The integrated EOMs can be solved for $\hat{a}_0'$ and $a_0'$,

$$\hat{a}_0' = \frac{n_B}{\sqrt{u^5 + n_B^2 + n_I^2}}, \qquad a_0' = \frac{n_I}{\sqrt{u^5 + n_B^2 + n_I^2}}, \tag{110}$$

which can be integrated once more to obtain the solutions

$$\hat{a}_0(u) = \mu_B - \frac{Cn_B}{(n_B^2 + n_I^2)^{3/10}} + \frac{n_B u}{\sqrt{n_B^2 + n_I^2}} \, {}_2F_1\left[\frac{1}{5}, \frac{1}{2}, \frac{6}{5}, -\frac{u^5}{n_B^2 + n_I^2}\right], \tag{111a}$$

$$a_0(u) = \mu_I - \frac{Cn_I}{(n_B^2 + n_I^2)^{3/10}} + \frac{n_I u}{\sqrt{n_B^2 + n_I^2}} \, {}_2F_1\left[\frac{1}{5}, \frac{1}{2}, \frac{6}{5}, -\frac{u^5}{n_B^2 + n_I^2}\right], \tag{111b}$$

together with the coupled equations

$$\mu_B = \frac{n_B}{\sqrt{n_B^2 + n_I^2}} \left\{ C(n_B^2 + n_I^2)^{1/5} - u_T \, {}_2F_1\left[\frac{1}{5}, \frac{1}{2}, \frac{6}{5}, -\frac{u_T^5}{n_B^2 + n_I^2}\right] \right\}, \tag{112a}$$

$$\mu_I = \frac{n_I}{\sqrt{n_B^2 + n_I^2}} \left\{ C(n_B^2 + n_I^2)^{1/5} - u_T \, {}_2F_1\left[\frac{1}{5}, \frac{1}{2}, \frac{6}{5}, -\frac{u_T^5}{n_B^2 + n_I^2}\right] \right\}, \tag{112b}$$

which relate the chemical potentials to the densities. Here ${}_2F_1$ is the hypergeometric function, and we have abbreviated $C \equiv \Gamma(3/10)\Gamma(6/5)/\sqrt{\pi}$. The renormalized free energy becomes

$$\Omega = \int_{u_T}^{\infty} du \, u^{5/2} \left( \frac{1}{\sqrt{1 + \frac{n_B^2}{u^5} + \frac{n_I^2}{u^5}}} - 1 \right) - \frac{2}{7} u_T^{7/2}$$

$$= -\frac{2}{7} u_T^{7/2} \, {}_2F_1\left[-\frac{7}{10}, \frac{1}{2}, \frac{3}{10}, -\frac{n_B^2 + n_I^2}{u_T^5}\right]. \tag{113}$$

In the zero-temperature limit, this reduces to the simple result

$$\Omega = -\frac{2(\mu_B^2 + \mu_I^2)^{7/4}}{7C^{5/2}}, \tag{114}$$

and the baryon and isospin densities are

$$n_B = \frac{\mu_B(\mu_B^2 + \mu_I^2)^{3/4}}{C^{5/2}}, \qquad n_I = \frac{\mu_I(\mu_B^2 + \mu_I^2)^{3/4}}{C^{5/2}}. \tag{115}$$

## 4.4 Results: deconfined geometry

Compared to the confined geometry, where we explored the phase structure systematically in Sec. 3, the deconfined geometry is expected to have a richer phase structure due to the nontrivial temperature dependence and the existence of the chirally symmetric phase. We



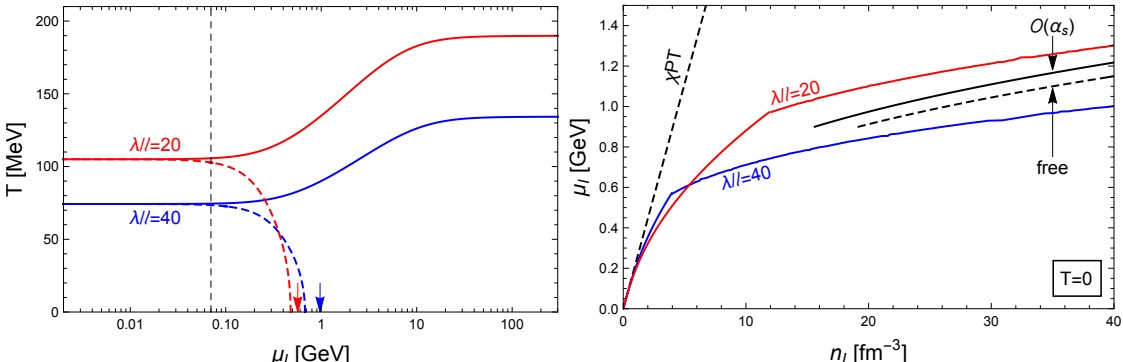

Figure 8: **Left panel:** Phase transition (solid) between the pion-condensed phase without baryons and the chirally symmetric phase in the plane of temperature $T$ and isospin chemical potential $\mu_I$ for $\mu_B = 0$ and two different values of $\lambda/\ell$. The arrows indicate the zero-temperature onset of baryons (in the presence of a pion condensate). The dashed curves show the corresponding chiral phase transition in the absence of pion condensation. The vertical thin dashed line marks the zero-temperature critical chemical potential for pion condensation if a physical pion mass was taken into account, $\mu_I = m_\pi/2$. **Right panel:** Isospin chemical potential as a function of the isospin density $n_I$ for $\mu_B = T = 0$ and the same values of $\lambda/\ell$ as in the left panel. The cusp in the curves corresponds to the baryon onset. The curves are compared to the ones from chiral perturbation theory ($\chi$PT), free quarks, and perturbative QCD to order $\alpha_s$.

leave a systematic study of the full phase diagram to the future and focus on a few key features which can be compared to known results from the literature. In particular, we shall only consider the case of vanishing baryon chemical potential. This case relates to various studies using lattice QCD [16–18, 20], perturbative QCD [22, 28, 29], chiral perturbation theory [21–27], and phenomenological models [83, 84].

Our results are shown in Fig. 8. To obtain the physical units in these plots we have, firstly, used the factors in table 1. Secondly, one finds that for a given $\lambda/\ell$ only a value of the energy scale $L^{-1}$ is needed (and not also of $M_{\mathrm{KK}}$) to obtain the results in the figure. We fix this scale by reproducing the physical pion decay constant $f_\pi \simeq 93\,\mathrm{MeV}$, with $f_\pi$ given in terms of the model parameters in Eq. (104). For the two choices in Fig. 8 we find $L^{-1} \simeq 480\,\mathrm{MeV}$ for $\lambda/\ell = 40$ and $L^{-1} \simeq 680\,\mathrm{MeV}$ for $\lambda/\ell = 20$. Since antipodal branes correspond to $\ell = \pi$, the geometric setup only makes sense for $\ell < \pi$, while the chirally broken phase in the deconfined geometry only exists for $\ell < 0.30768\pi$ [85], and our decompactified limit even requires $\ell \ll \pi$ (although one might consider the setup as an effective approach, which is then extrapolated beyond its original regime of validity). Let us use the critical value $\ell \simeq 0.3\pi$ for a rough comparison. For $\lambda/\ell = 40$ this implies $\lambda \simeq 38$ and $M_{\mathrm{KK}} \simeq 450\,\mathrm{MeV}$, while for $\lambda/\ell = 20$ we have $\lambda \simeq 19$ and $M_{\mathrm{KK}} \simeq 640\,\mathrm{MeV}$. The original fit by Sakai and Sugimoto [32], using the pion decay constant and the rho meson mass (however in the confined geometry), gave $\lambda \simeq 17$ and $M_{\mathrm{KK}} \simeq 949\,\mathrm{MeV}$. Compared to these values our result for $\lambda/\ell = 20$ seems more sensible, although even in this case our Kaluza-Klein scale is somewhat low. Nevertheless, the two different values for $\lambda/\ell$ are useful to observe a tendency of our results upon variation of the 't Hooft coupling.

The left panel of Fig. 8 shows the chiral phase transition in the $T$-$\mu_I$ plane in the absence of baryons. Since we work in the chiral limit, pions condense for any $\mu_I$ at sufficiently small $T$, and thus for all $\mu_I$ the solid curve separates the pion-condensed phase from the chirally symmetric phase. In a more realistic scenario, where the pion mass is nonzero, there is no pion condensation for small $\mu_I$. Therefore, in this regime, the chiral phase transition will be

given by the dashed curve, for which we have ignored pion condensation (this curve will also be slightly corrected by a quark mass term [60]). We have indicated the value of the chemical potential $\mu_I = m_\pi/2$, where we expect the zero-temperature onset of pion condensation once quark masses are included. Then, for larger values of $\mu_I$ we expect the phase transition line to approach our solid curve, where pion condensation is taken into account. This suggests a picture not unlike the recent results from lattice QCD [20]. Differences are location and nature of our $\mu_I = 0$ transition, which occurs at a smaller temperature than in the real world (although our value strongly depends on the choice of $\lambda/\ell$) and is of first order, as in many related holographic studies, but in contrast to the smooth crossover in QCD. Also, we observe no chiral restoration for small temperatures as $\mu_I$ is increased. Instead, we find that the critical temperature saturates at a value almost twice as large as the critical temperature at $\mu_I = 0$. This curve is obtained without taking into account baryonic matter for simplicity and also without taking into account any additional meson condensation, for instance condensation of rho mesons, which we have ignored throughout the paper. We have indicated the zero-temperature baryon onset, which occurs at $\mu_I \simeq 970\,\text{MeV} \simeq 6.9 m_\pi$ for $\lambda/\ell = 20$. A complete nonzero-temperature study is left for the future.

The zero-temperature effect of the baryons is illustrated in the right panel. We see that the relation between isospin density and chemical potential follows chiral perturbation theory for small $\mu_I$, as already observed analytically, see Eq. (103) (the expressions in the parentheses in that equation are exactly the dimensionful quantities plotted in Fig 8, denoted for simplicity by the same symbols as their dimensionless counterparts). Then, a deviation from chiral perturbation theory is observed already before baryons appear through a second-order onset. Such a second-order onset was also seen in the confined geometry at $\mu_B = 0$, as discussed in the context of Fig. 3. Recall from the evaluation in the confined geometry that after this onset there is a coexistence between two states with positive and negative net baryon number, see Secs. 3.1 and 3.2. The same qualitative behavior is found in the deconfined geometry. Therefore, we may think of the isospin density shown here to receive contributions from baryons (net positive baryon number) "just above" the phase transition at $\mu_B = 0$ or, equivalently, contributions from antibaryons (net negative baryon number) "just below" the phase transition. We see that baryons create a further deviation from the results of chiral perturbation theory, rendering the isospin density more sensitive to changes in the isospin chemical potential. The figure suggests that this deviation is required to approach the limit at asymptotically large $\mu_I$. For comparison we have plotted the result of free massless two-flavor quark matter (since $\mu_B = 0$ here, this is a gas of up and anti-down quarks) and the correction to linear order in the strong coupling constant $\alpha_s$ using the running of the coupling from Ref. [86]. Since our model is not asymptotically free, we do not expect our curves to reproduce these weak-coupling results. It is nevertheless intriguing that our result seems to roughly interpolate between chiral perturbation theory and the ultra-high density regime. In this sense, our holographic model behaves similarly to the lattice results of Ref. [18], which however are obtained at $n_B = 0$, see also Fig. 3 in Ref. [86] (where an unphysically large pion mass is assumed).

## 5 Summary and outlook

We have studied spatially uniform baryonic matter in the presence of an isospin asymmetry within the Witten-Sakai-Sugimoto model. Baryon number is created by a homogeneous ansatz for the spatial components of the non-abelian part of the $U(2)$ gauge field in the bulk, following earlier studies for isospin-symmetric baryonic matter. An isospin chemical potential $\mu_I$ gives rise to a non-trivial profile for the temporal non-abelian component $a_0$ and deforms the baryonic field together with the abelian gauge potential $\hat{a}_0$ associated with the baryon chemical potential $\mu_B$. We have also allowed for a pion condensate to coexist with baryonic matter

and have compared the free energies of the various possible phases. This has been done in the confined geometry – best suited for the introduction of our concepts and a complete evaluation – and in the deconfined geometry (within the decompactified limit) – which is more difficult, but better suited for a comparison to real-world QCD due to the existence of a chiral phase transition and a nontrivial temperature dependence.

We have found that the phase of coexistence between pion condensation and baryonic matter plays a very prominent role in the phase diagram. In the confined geometry we have shown that within our approximations, most notably neglecting the pion mass, this coexistence phase is energetically preferred in the entire $\mu_B$-$\mu_I$ plane except for a corner of sufficiently small $\mu_B$ and $\mu_I$, where baryons cannot be created and the pure pion-condensed phase is preferred. In particular, even at $\mu_B = 0$ baryons (or, equivalently, a mirror state with anti-baryons) are created for sufficiently large $\mu_I$. Even though our approximation is expected to be valid only at large baryon densities, we have pointed out that if the baryon density is taken to zero with $\mu_B$ held fixed the system approaches a certain finite value of $\mu_I$, which we can interpret as a baryon mass. This is different at fixed $\mu_I$, where $\mu_B$ diverges as the baryon density goes to zero, a known shortcoming of the approximation. We have also discussed charge neutral, beta-equilibrated baryonic matter, having in mind future applications to the physics of compact stars, and computed the trajectory of this matter in our phase diagram. Most strikingly, we have found an extremely large proton fraction very close to isospin-symmetric matter, in contrast to realistic nuclear matter where, at least at not too large densities, the proton fraction is about 10% or lower. This result is related to an unphysically large symmetry energy, which can be explained by the continuous isospin spectrum, a large-$N_c$ artifact due to our semi-classical approximation without quantization of the holographic baryonic states.

Using the deconfined geometry, we have pointed out that the model can also be used for predictions regarding the phase structure in the $T$-$\mu_I$ plane at $\mu_B = 0$. In the absence of baryons, we have computed the critical temperature for the chiral phase transition, which in the given model saturates at large $\mu_I$. For zero temperature we have demonstrated that the isospin density agrees with chiral perturbation theory for small $\mu_I$ and deviates at large $\mu_I$ – within the pion-condensed phase and at even larger $\mu_I$ due to the appearance of baryons – in a way that is qualitatively the same as suggested by lattice QCD and by perturbative benchmarks at asymptotically large $\mu_I$. In particular the appearance of baryons is an interesting prediction that should be investigated further in different approaches, possibly using lattice gauge theory.

Our study is the first to include isospin-asymmetric baryonic matter in a consistent way within a holographic model and thus various improvements are necessary for a more realistic approach, be it in the Witten-Sakai-Sugimoto model or in a different holographic setup. Firstly, we have only started to evaluate our setup in the deconfined geometry, and a more systematic study, although numerically somewhat challenging, can be done with the present approach, for instance regarding the effect of temperature on asymmetric baryonic matter. No further approximation would be required and our model consistently accounts for pions and baryons and their interactions at any temperature. This may be of relevance in the context of core-collapse supernovae or neutron star mergers, where the potential importance of thermal pions was pointed out recently [87]. More conceptual work is needed to connect our current approach with the instanton solutions for single baryons and the various many-baryon approximations based on these solutions. This is probably necessary to account for baryonic matter made of neutrons and protons rather than a continuum of isospin states. More straightforwardly, one can include a nonzero pion mass into our approach, which will affect the physics at not too large $\mu_I$ (relevant for compact stars) and which can be done along the lines of Refs. [12, 60]. Other possible extensions include the addition of a magnetic field, which has been done in similar calculations [53, 88] and which could be compared to results on the lattice [89], and the question of isospin-asymmetric quarkyonic matter, building on the symmetric case [12]

and comparing the results to a recently developed phenomenological approach [90].

## Acknowledgements

We thank Gergely Endrődi, Eduardo Fraga, and Norberto Scoccola for helpful discussions. This work is supported by the Leverhulme Trust under grant no RPG-2018-153. A.S. acknowledges support by the Science & Technology Facilities Council (STFC) in the form of an Ernest Rutherford Fellowship. The work of N.K. is supported by the ERC Consolidator Grant 772408-Stringlandscape.

## A  Single-instanton solution with isospin (deconfined geometry)

In this appendix we derive the effect of the isospin chemical potential on the single-instanton configuration. We present the details of the calculation for the deconfined geometry, for the confined geometry one proceeds analogously. Here we restrict ourselves to a single baryon in the vacuum, not in the presence of a pion condensate.

The first part of the derivation follows appendix A of Ref. [10]. We start from the YM approximation, by expanding the DBI action (73) up to second order in the field strengths, which, together with the CS term gives

$$S \simeq S_0 + S_{\text{YM}} + S_{\text{CS}}, \tag{116}$$

where $S_0$ is a purely geometric term, independent of the field strengths,

$$S_0 = \frac{N_f \mathcal{N}}{M_{\text{KK}}^3 T} \int d^3x \int_{u_c}^{\infty} du\, u^{5/2} \sqrt{1 + u^3 f_T x_4'^2}, \tag{117}$$

where the YM action is

$$S_{\text{YM}} = \frac{\mathcal{N}}{2\lambda_0^2 M_{\text{KK}}^3 T} \int d^3x \int_{u_c}^{\infty} du\, u^{5/2} \left\{ \frac{\lambda_0^2 \text{Tr}[\mathcal{F}_{0z}^2] + f_T \text{Tr}[\mathcal{F}_{iz}^2]}{\sqrt{1 + u^3 f_T x_4'^2}} \right.$$

$$\left. + \frac{\sqrt{1 + u^3 f_T x_4'^2}}{u^3 f_T} \left( \lambda_0^2 \text{Tr}[\mathcal{F}_{0i}^2] + \frac{f_T}{2} \text{Tr}[\mathcal{F}_{ij}^2] \right) \right\}, \tag{118}$$

and where the CS contribution comes from the first term of the general form (9),

$$S_{\text{CS}} = -i\frac{3\mathcal{N}}{2\lambda_0^2 M_{\text{KK}}^3 T} \int d^3x \int_{u_c}^{\infty} du\, \hat{a}_0 \text{Tr}[F_{iu} F_{jk}] \epsilon_{ijk}. \tag{119}$$

In the absence of baryons (or other sources) $S_0$ yields the vacuum solution for the embedding $x_4(u)$, namely Eq. (96). At low energy, a single baryon is created at $u = u_c$, i.e., at $z = 0$, with a width that goes to zero for $\lambda \to \infty$. We will thus use the (temperature-dependent) embedding given by Eq. (96) with $u_c$ computed from Eq. (85a) (without backreaction of the single baryon on this embedding, such that $S_0$ can be ignored from now on), and will apply an expansion in powers of $z$, which is equivalent to a strong coupling expansion. The leading term is of order $\lambda$ and receives a contribution only from the YM term,

$$S_{\text{YM}}^{(1)} = \frac{\mathcal{N}}{4\lambda_0^2 M_{\text{KK}}^3 T} \frac{u_c \sqrt{f_T(u_c)}}{\gamma} \int d^3x \int_{-\infty}^{\infty} dz \left( \frac{1}{2}\text{Tr}[F_{ij}^2] + \gamma^2 \text{Tr}[F_{iz}^2] \right), \tag{120}$$

where only the non-abelian field strengths contribute (recall the decomposition (7)), and where we have abbreviated

$$\gamma^2 \equiv 6u_c^3 \left(1 - \frac{5u_T^3}{8u_c^3}\right). \tag{121}$$

From the action (120) we derive the EOMs for the non-abelian gauge fields,

$$\partial_j F_{ji}^a - 2\epsilon_{abc} a_j^b F_{ji}^c = \gamma^2 (\partial_z F_{iz}^a - 2\epsilon_{abc} a_z^b F_{iz}^c), \tag{122a}$$

$$\partial_i F_{iz}^a - 2\epsilon_{abc} a_i^b F_{iz}^c = 0, \tag{122b}$$

which are solved by the BPST instanton solutions

$$a_z^a(\boldsymbol{x}, z) = -\frac{1}{\gamma} \frac{x_a}{\xi^2 + (\rho/\gamma)^2}, \qquad a_i^a(\boldsymbol{x}, z) = \frac{z/\gamma \, \delta_{ia} - \epsilon_{ija} x_j}{\xi^2 + (\rho/\gamma)^2}, \tag{123}$$

with the width $\rho$, to be determined dynamically in the presence of the subleading terms, and $\xi^2 \equiv x^2 + (z/\gamma)^2$. The corresponding field strengths are

$$F_{zi}^a(\boldsymbol{x}, z) = \frac{2(\rho/\gamma)^2 \delta_{ia}}{\gamma[\xi^2 + (\rho/\gamma)^2]^2}, \qquad F_{ij}^a(\boldsymbol{x}, z) = \frac{2(\rho/\gamma)^2 \epsilon_{ija}}{[\xi^2 + (\rho/\gamma)^2]^2}. \tag{124}$$

For the temporal components (both abelian and non-abelian) we need to compute the subleading contributions of order $\lambda^0$. At this order we have contributions form the CS term (119) and from the subleading YM term,

$$S_{\text{YM}}^{(0)} = \frac{\mathcal{N}}{4\lambda_0^2 M_{\text{KK}}^3 T} \frac{u_c \sqrt{f_T(u_c)}}{\gamma} \int d^3 x \int_{-\infty}^{\infty} dz \left[ \lambda_0^2 \frac{\text{Tr}[\mathcal{F}_{0i}^2] + \gamma^2 \text{Tr}[\mathcal{F}_{0z}^2]}{f_T(u_c)} + 3u_c z^2 \text{Tr}[\mathcal{F}_{iz}^2] \right.$$

$$\left. + \frac{4u_c^6 + 10u_c^3 u_T^3 - 5u_T^6}{8\gamma^2 u_c^5 f_T(u_c)} z^2 \left( \frac{\text{Tr}[\mathcal{F}_{ij}^2]}{2} + \gamma^2 \text{Tr}[\mathcal{F}_{iz}^2] \right) \right]. \tag{125}$$

The resulting EOMs for $\hat{a}_0$ and $a_0$ are thus

$$\partial_i \hat{F}_{i0} + \gamma^2 \partial_z \hat{F}_{z0} = -i \frac{3\gamma \sqrt{f_T(u_c)}}{2\lambda_0^2 u_c} F_{zi}^a F_{jk}^a \epsilon_{ijk}, \tag{126a}$$

$$\partial_i F_{i0}^a - 2\epsilon_{abc} a_i^b F_{i0}^c = \gamma^2 (\partial_z F_{0z}^a - 2\epsilon_{abc} a_z^b F_{0z}^c). \tag{126b}$$

We impose the following boundary conditions for the Euclidean fields (i.e., after $\hat{a}_0 \to i\hat{a}_0$ and $a_0 \to ia_0$)

$$\hat{a}_0(z \to \pm\infty) = \mu_B, \qquad a_0^a(z \to \pm\infty) = v^a. \tag{127}$$

These are the $\sigma$-type boundary conditions explained in the main text since here we do not take into account pion condensation. We have also used a general three-vector $\boldsymbol{v}$ for the boundary values of the non-abelian part, the isospin chemical potential is then introduced by $\boldsymbol{v} = (0, 0, \mu_I)$ or, equivalently, simply by $v = |\boldsymbol{v}| = \mu_I$. With these boundary conditions the EOMs are solved by the Euclidean temporal components

$$\hat{a}_0(\boldsymbol{x}, z) = \mu_B - \frac{3\sqrt{f_T(u_c)}}{2\lambda_0^2 u_c} \frac{\xi^2 + 2(\rho/\gamma)^2}{[\xi^2 + (\rho/\gamma)^2]^2},$$

$$a_0^a(\boldsymbol{x}, z) = \frac{[(z/\gamma)^2 - x^2] v_a + 2x_a \boldsymbol{v} \cdot \boldsymbol{x} - 2(z/\gamma)\epsilon_{abc} v_b x_c}{\xi^2 + (\rho/\gamma)^2}, \tag{128}$$

which leads to the field strengths

$$F_{0z}^a = -\frac{2(\rho/\gamma)^2[v_a(z/\gamma) - \epsilon_{abc}v_b x_c]}{\gamma[\xi^2 + (\rho/\gamma)^2]^2},$$

$$F_{0i}^a = \frac{2(\rho/\gamma)^2[-\delta_{ia}\boldsymbol{v}\cdot\boldsymbol{x} + (z/\gamma)\epsilon_{iab}v_b + v_a x_i - v_i x_a]}{[\xi^2 + (\rho/\gamma)^2]^2}. \tag{129}$$

Reinserting all solutions into the action $S \simeq S_{\text{YM}}^{(1)} + S_{\text{YM}}^{(0)} + S_{\text{CS}}$ and performing the $\boldsymbol{x}$ and $z$ integrals yields the free energy $TS = \lambda_0 N_c M_{\text{KK}}\phi$, with the dimensionless free energy

$$\phi = \frac{u_c\sqrt{f_T(u_c)}}{3}\left[1 + \frac{9\gamma^2}{5\rho^2\lambda_0^2 u_c^2} + \frac{u_c\beta\rho^2}{\gamma^2 f_T(u_c)}\right] - \mu_B, \tag{130}$$

where we have abbreviated

$$\beta \equiv \beta_0 - \frac{\lambda_0^2 v^2}{u_c}, \qquad \beta_0 \equiv 1 - \frac{u_T^3}{8u_c^3} - \frac{5u_T^6}{16u_c^6}. \tag{131}$$

For $v = 0$ we have $\beta = \beta_0$ and we recover the result of Ref. [10]. We see that the result only depends on the modulus of $\boldsymbol{v}$ (and not on the $SU(2)$ direction) and will identify $v = \mu_I$ from now on. The minimization with respect to the instanton width $\rho$ yields

$$\rho^2 = \frac{12\pi}{\sqrt{5}\lambda}\frac{\gamma^2\sqrt{f_T(u_c)}}{u_c^{3/2}\beta^{1/2}}. \tag{132}$$

As expected, we have found $\rho \sim 1/\sqrt{\lambda}$, which justifies the expansion above *a posteriori*. Moreover, we find that the width is increased by the presence of the isospin chemical potential, and for $\rho^2$ to be real we need $\beta > 0$, which imposes the constraint

$$|\mu_I| < \frac{\sqrt{u_c\beta_0}}{\lambda_0}. \tag{133}$$

By using (132) the free energy at the stationary point can be written as

$$\phi = \frac{u_c\sqrt{f_T(u_c)}}{3}\left[1 + \frac{6\beta^{1/2}}{\sqrt{5}\lambda_0 u_c^{1/2}\sqrt{f_T(u_c)}}\right] - \mu_B, \tag{134}$$

so that the baryon and isospin numbers are

$$N_B = -\frac{\partial\phi}{\partial\mu_B} = 1, \qquad N_I = -\frac{\partial\phi}{\partial\mu_I} = \frac{2}{\sqrt{5}}\frac{\mu_I\lambda_0}{u_c^{1/2}\beta^{1/2}}. \tag{135}$$

As it should be, the baryon number is 1, according to the winding number of the instanton solution, while $N_I$ monotonically increases with $\mu_I$. Despite the upper limit for $\mu_I$ (133), arbitrarily large values of $N_I$ can be assumed. In other words, with $\mu_I$ we can tune the isospin content of a single baryon continuously in the entire range $N_I \in [-\infty, \infty]$.

We can also compute the (dimensionless) energy $e$ of a single baryon via the relation $\phi = e - \mu_I N_I - \mu_B N_B$, which yields

$$e = \frac{u_c\sqrt{f_T(u_c)}}{3}\left[1 + \frac{6\beta_0}{\sqrt{5}\lambda_0 u_c^{1/2}\beta^{1/2}\sqrt{f_T(u_c)}}\right]. \tag{136}$$

This shows that the mass of the single baryon increases monotonically (and without limit) with $\mu_I$. With Eq. (135) we can derive the useful relation

$$\frac{\beta_0}{\beta} = 1 + \frac{5}{4}N_I^2, \tag{137}$$

such that the baryon mass expressed in terms of the isospin number is Eq. (75) in the main text.

# B  Symmetrized trace

Here we apply the symmetrized trace prescription of Ref. [78] to the non-abelian DBI action in the presence of the isospin chemical potential. The idea is to first compute the determinant as if the field-strengths were abelian, which was already done in the main text, see Eq. (74). We then expand the square root, still ignoring the non-abelian structure, and finally take the so-called symmetrized trace of each term in the expansion, i.e., we sum over all possible permutations of the field strengths before taking the usual trace.

Within our homogeneous ansatz we can write the DBI Lagrangian (74) as

$$
\mathcal{L}_{\mathrm{DBI}} = u^{5/2} \operatorname{STr}\sqrt{t + t_a \sigma_a + t_{ab}\sigma_a\sigma_b + t_{abc}\sigma_a\sigma_b\sigma_c + t_{abcd}\sigma_a\sigma_b\sigma_c\sigma_d}, \quad (138)
$$

where we have abbreviated

$$
t = 1 + u^3 f_T x_4'^2 - \hat{a}_0'^2, \quad (139a)
$$

$$
t_a = -2\hat{a}_0' a_0' \delta_{a3}, \quad (139b)
$$

$$
t_{ab} = (g_1 + t g_2)\frac{\delta_{ab}}{3} - \frac{g_3}{2}(1 + u^3 f_T x_4'^2)(\delta_{ab} - \delta_{a3}\delta_{b3}) - \delta_{a3}\delta_{b3}a_0'^2, \quad (139c)
$$

$$
t_{abc} = -\hat{a}_0'\left[\frac{2g_2}{3}a_0'\delta_{a3}\delta_{bc} - \frac{\lambda_0^2 h^3 h' a_0}{2u^3}(\delta_{ac}\delta_{b3} - \delta_{ab}\delta_{c3})\right], \quad (139d)
$$

$$
t_{abcd} = -\frac{g_2}{3}a_0'^2\delta_{a3}\delta_{b3}\delta_{cd} + \frac{g_1 g_2}{9}\delta_{ab}\delta_{cd} - \frac{g_1 g_3}{6}(\delta_{ab} - \delta_{a3}\delta_{b3})\delta_{cd}
$$

$$
+ \epsilon_{ab3}\epsilon_{cd3}\frac{g_3}{6}\left[g_1 - g_2(1 + u^3 f_T x_4'^2)\right] - \frac{\lambda_0^2 h^3 h' a_0 a_0'}{2u^3}\delta_{a3}(\delta_{bd}\delta_{c3} - \delta_{bc}\delta_{d3}), \quad (139e)
$$

with $g_1$, $g_2$, $g_3$ defined in Eq. (79).

In the isospin-symmetric case we have $a_0 = a_0' = g_3 = 0$, and the square root of the determinant becomes a function of the variable $x = \boldsymbol{\sigma}^2 = \sigma_a\sigma_a$,

$$
\mathcal{L}_{\mathrm{DBI}} = u^{5/2}\operatorname{STr}[f(x)], \qquad f(x) = \sqrt{\left(t + \frac{g_1}{3}x\right)\left(1 + \frac{g_2}{3}x\right)}. \quad (140)
$$

For this case, the symmetrized trace was computed in Ref. [10], following appendix A in Ref. [79]. We can write the formal expansion as

$$
f(x) = \sum_{n=0}^{\infty} c_n x^n, \quad (141)
$$

with coefficients $c_n$, and introduce the notation

$$
[(\boldsymbol{\sigma}^2)^n]_{\mathrm{sym}} \equiv \frac{1}{N(n)}\sum_{i=1}^{N(n)} \pi_i[(\boldsymbol{\sigma}^2)^n], \quad (142)
$$

where each $\pi_i$ is a permutation of the $2n$ Pauli matrices which are pairwise contracted. The sum is only over distinct permutations, and the number of distinct permutations is denoted by $N(n)$. For instance, for $n = 2$ we have $N(2) = 3$, and the permutations are $\sigma_a\sigma_a\sigma_b\sigma_b$, $\sigma_a\sigma_b\sigma_a\sigma_b$, $\sigma_a\sigma_b\sigma_b\sigma_a$. For general $n$ one easily finds

$$
N(n) = (2n-1)!! = \frac{(2n)!}{2^n n!}. \quad (143)
$$

With the help of induction one proves [79]

$$[(\boldsymbol{\sigma}^2)^n]_{\text{sym}} = (2n+1)\mathbb{1}, \tag{144}$$

which implies

$$\text{STr}[f(x)] = \sum_{n=0}^{\infty} c_n \text{STr}[(\boldsymbol{\sigma}^2)^n] = 2\sum_{n=0}^{\infty} c_n(2n+1) = \left(4x\frac{\partial}{\partial x} + 2\right)f(x)\Big|_{x=1}, \tag{145}$$

such that one can actually avoid performing the series expansion explicitly.

The presence of a non-zero isospin chemical potential makes this analysis more complicated. In this case, the square root of the determinant gives rise to a function of the variables $x = \boldsymbol{\sigma}^2$ and $y = \sigma_3$, whose explicit form can be obtained from Eqs. (138) and (139), and whose formal expansion we can write as

$$f(x,y) = \sum_{n,p=0}^{\infty} c_{n,p} x^n y^p. \tag{146}$$

We need to compute the symmetrized trace of $f(x,y)$. It is clear that the terms with odd $p$ give a vanishing contribution. For even $p$ we need to generalize Eq. (144). Setting $p = 2m$ we will show that the following identity holds,

$$[(\boldsymbol{\sigma}^2)^n(\sigma_3^2)^m]_{\text{sym}} = \frac{2(n+m)+1}{2m+1}\mathbb{1}. \tag{147}$$

This is proven by induction in $n$ and $m$. For $m = 0$ we simply reproduce Eq. (144), while for $n = 0$ Eq. (147) holds trivially. Therefore, it is enough to show that, for a given pair $(n,m)$, Eq. (147) is a consequence of the $(n-1,m)$ and $(n,m-1)$ cases. To show this, let us first denote the number of distinct permutations of $(\boldsymbol{\sigma}^2)^n(\sigma_3^2)^m$ by $N(n,m)$. One finds

$$N(n,m) = (2n-1)!!\frac{(2n+2m)!}{(2n)!(2m)!} = \frac{(2n+2m)!}{2^n n!(2m)!}. \tag{148}$$

We can divide the sum over all permutations into categories depending on the first two matrices as follows,

$$N(n,m)[(\boldsymbol{\sigma}^2)^n(\sigma_3^2)^m]_{\text{sym}} = \sum_{i=1}^{N(n,m)} \pi_i[(\boldsymbol{\sigma}^2)^n(\sigma_3^2)^m]$$

$$= \sigma_a\sigma_a \sum_{i=1}^{N_1} \pi_i[(\boldsymbol{\sigma}^2)^{n-1}(\sigma_3^2)^m] + \sigma_a\sigma_b \sum_{i=1}^{N_2} \pi_i[(\sigma_a\sigma_b\boldsymbol{\sigma}^2)^{n-2}(\sigma_3^2)^m]$$

$$+ (\sigma_a\sigma_3 + \sigma_3\sigma_a)\sum_{i=1}^{N_3} \pi_i[\sigma_a\sigma_3(\boldsymbol{\sigma}^2)^{n-1}(\sigma_3^2)^{m-1}] + \sigma_3\sigma_3 \sum_{i=1}^{N_4} \pi_i[(\boldsymbol{\sigma}^2)^n(\sigma_3^2)^{m-1}], \tag{149}$$

where

$$N_1 = N(n-1,m),\ N_2 = (2n-2)N(n-1,m),\ N_3 = 2mN(n-1,m),\ N_4 = N(n,m-1). \tag{150}$$

Consequently, with $\{\sigma_a, \sigma_b\} = 2\delta_{ab}$,

$$
\begin{aligned}
N(n,m)[(\boldsymbol{\sigma}^2)^n(\sigma_3^2)^m]_{\text{sym}} &= 3N_1[(\boldsymbol{\sigma}^2)^{n-1}(\sigma_3^2)^m]_{\text{sym}} + N_2\delta_{ab}[\sigma_a\sigma_b(\boldsymbol{\sigma}^2)^{n-2}(\sigma_3^2)^m]_{\text{sym}} \\
&\quad + 2N_3\delta_{a3}[\sigma_a\sigma_3(\boldsymbol{\sigma}^2)^{n-1}(\sigma_3^2)^{m-1}]_{\text{sym}} + N_4[(\boldsymbol{\sigma}^2)^n(\sigma_3^2)^{m-1}]_{\text{sym}} \\
&= (3N_1 + N_2 + 2N_3)[(\boldsymbol{\sigma}^2)^{n-1}(\sigma_3^2)^m]_{\text{sym}} + N_4[(\boldsymbol{\sigma}^2)^n(\sigma_3^2)^{m-1}]_{\text{sym}} \\
&= N(n,m)\frac{2(n+m)+1}{2m+1}\mathbb{1},
\end{aligned}
\tag{151}
$$

where, in the last step, we have inserted Eq. (150) and used Eq. (147) for the $(n-1,m)$ and $(n, m-1)$ cases. This proves Eq. (147).

Thus, with Eq. (146) we find

$$
\text{STr}[f(x,y)] = 2\sum_{n,m} c_{n,2m} \frac{2(n+m)+1}{2m+1}.
\tag{152}
$$

This result cannot simply be written with the help of derivatives acting on $f(x,y)$ as in Eq. (145). Instead, we can introduce an integro-differential operator acting on $f(x, \tilde{y}w)$,

$$
\text{STr}[f(x,y)] = \left(2x\frac{\partial}{\partial x} + \tilde{y}\frac{\partial}{\partial \tilde{y}} + 1\right) \int_{-1}^{1} dw\, f(x, \tilde{y}w)\Big|_{x=\tilde{y}=1}.
\tag{153}
$$

Besides generating the denominator in (152), the auxiliary integral in $w$ removes the terms with odd powers of $y$ from (146) as needed. If $f$ is independent of $y$ this integration simply gives a factor 2, and we recover Eq. (145), as it should be.

This result can now in principle be used to compute the symmetrized trace in the presence of an isospin asymmetry (and within our homogeneous ansatz). However, the resulting expression is very lengthy and not particularly illuminating, and thus we do not include it here. For our main calculation in the deconfined geometry we apply the approximation (78), correct to $\mathcal{O}(F^2)$, for the reasons explained in the main text.

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
