# Peer review of "Isospin asymmetry in holographic baryonic matter"

_SciPost Physics, doi:SciPost Phys. 11, 029 (2021)_

## Round 2 · Author Response

We thank both referees for their positive report and referee 2 for the constructive questions and comments. Our replies to her/his points are as follows (in the same order as in the report)

(1) Our strategy was, firstly, to discuss the simplest possible version of the model (Secs 2 & 3: confined geometry, antipodal separation, YM approximation) in order to focus on a transparent discussion of the main conceptual issues. Secondly, we used the "most realistic" version (Sec 4, deconfined geometry, allowing for temperature effects and chiral symmetry restoration). The referee's suggestion, a square root-type action analogous to that of Eq. (78) in the context of the confined geometry, would be something "in between", computationally challenging but probably of limited value for real-world QCD. This would clearly be possible and the large-density behavior would be somewhat different, but we do not expect the main results to change substantially compared to what we found in Sec 4, at least in the interesting regimes of the phase transitions to baryonic matter. This expectation is based on our observations in the deconfined geometry, where we do use the square-root action and the phase diagram in the $\mu_B$-$\mu_I$ plane (not shown in the paper) is qualitatively the same.

(2) We agree with the referee that the explanations in Sec 2.4 are relatively brief, but we think that the main idea comes across. Therefore, and given that this is merely a recapitulation of the arguments given in Ref [52], we have decided to keep this section as it is.

(3) We have modified the sentence as follows, to make the statement clearer: "Most importantly, the energy differences between baryonic states with different isospin values go to zero as $N_c\to \infty$, such that the spectrum becomes continuous with respect to isospin."

(4) We thank the referee for this question, there is indeed a slight inaccuracy in our statement. As we explain in the manuscript, the problem about working with an expansion in powers of the field-strengths is that one does not get rid of the square roots in the Lagrangian due to the presence of the embedding function $x_4(u)$ (in which we obviously cannot expand). This problem, together with the higher-order terms in $F^2$, leads to equations of motion more complicated than Eqs. (81). In general, they do not allow one to solve algebraically for the derivatives of $x_4$ and $\hat{a}_0$ in a simple way. One would then have to implement a fully numerical evaluation. Now, as suggested by the referee, truncating at order $F^2$, i.e., using the YM Lagrangian, does lead to a set of equations of motion which can be solved for the derivatives of $x_4$ and $\hat{a}_0$ algebraically. In this sense, our statement regarding the "purely numerical evaluation" is not accurate. However, the actual solutions obtained in this way are much more complicated than those presented in the manuscript.

To be more precise, we have deleted the part about the "purely numerical evaluation" in the sentence the referee refers to, which now reads,

"Truncations of the resulting infinite series at ${\cal O}(F^2)$ or ${\cal O}(F^4)$ are possible, but also lead to a relatively complicated action due to the presence of the embedding function",

and we have changed the last sentence of this section to

"Had we used the YM approximation, which can be obtained by expanding the square root in Eq. (78) to second order in the field strengths, the resulting expressions would have been much more complicated."

(5) Yes, the referee is correct that this connection was already made in Ref [50], and we agree that our quote is not precise. We actually had in mind the analogue of Eq (92), which can be found in Ref [9]. We have now changed the sentence to

"..., was already pointed out in Ref. [50], see also Eq. (73) in Ref. [9]."

(6) We thank the referee for pointing this out. We have replaced "a_0(\infty)" by "$a_0(\infty) = \mu_I$".

(7) We have made several changes in the formulations in the first part of the paragraph that includes the sentence mentioned by the referee. We think the new version is clearer.

---

## Editorial Decision

published